# Effects of Therapeutic Platelet-Rich Plasma on Overactive Bladder via Modulating Hyaluronan Synthesis in Ovariectomized Rat

**DOI:** 10.3390/ijms24098242

**Published:** 2023-05-04

**Authors:** Jian-He Lu, Kuang-Shun Chueh, Tai-Jui Juan, Jing-Wen Mao, Rong-Jyh Lin, Yi-Chen Lee, Mei-Chen Shen, Ting-Wei Sun, Hung-Yu Lin, Yung-Shun Juan

**Affiliations:** 1Center for Agricultural, Forestry, Fishery, Livestock and Aquaculture Carbon Emission Inventory and Emerging Compounds, General Research Service Center, National Pingtung University of Science and Technology, Pingtung 912301, Taiwan; 2Graduate Institute of Clinical Medicine, College of Medicine, Kaohsiung Medical University, Kaohsiung 807378, Taiwan; 3Department of Urology, Kaohsiung Municipal Ta-Tung Hospital, Kaohsiung 801735, Taiwan; 4Department of Urology, Kaohsiung Medical University Hospital, Kaohsiung 807378, Taiwan; 5Department of Medicine, National Defense Medical College, Taipei 114201, Taiwan; 6Department of Parasitology, School of Medicine, College of Medicine, Kaohsiung Medical University, Kaohsiung 807378, Taiwan; 7Graduate Institute of Medicine, College of Medicine, Kaohsiung Medical University, Kaohsiung 807378, Taiwan; 8Center for Tropical Medicine and Infectious Disease Research, Kaohsiung Medical University, Kaohsiung 807378, Taiwan; 9Department of Anatomy, School of Medicine, College of Medicine, Kaohsiung Medical University, Kaohsiung 807378, Taiwan; 10Department of Urology, College of Medicine, Kaohsiung Medical University, Kaohsiung 807378, Taiwan; 11School of Medicine, College of Medicine, I-Shou University, Kaohsiung 824005, Taiwan; 12Division of Urology, Department of Surgery, E-Da Cancer Hospital, I-Shou University, Kaohsiung 840301, Taiwan; 13Regenerative Medicine and Cell Therapy Research Cancer, Kaohsiung Medical University, Kaohsiung 807378, Taiwan

**Keywords:** overactive bladder, ovariectomy, platelet rich plasma, hyaluronan

## Abstract

Postmenopausal women who have ovary hormone deficiency (OHD) may experience urological dysfunctions, such as overactive bladder (OAB) symptoms. This study used a female Sprague Dawley rat model that underwent bilateral ovariectomy (OVX) to simulate post-menopause in humans. The rats were treated with platelet-rich plasma (PRP) or platelet-poor plasma (PPP) after 12 months of OVX to investigate the therapeutic effects of PRP on OHD-induced OAB. The OVX-treated rats exhibited a decrease in the expression of urothelial barrier-associated proteins, altered hyaluronic acid (hyaluronan; HA) production, and exacerbated bladder pathological damage and interstitial fibrosis through NFƘB/COX-2 signaling pathways, which may contribute to OAB. In contrast, PRP instillation for four weeks regulated the inflammatory fibrotic biosynthesis, promoted cell proliferation and matrix synthesis of stroma, enhanced mucosal regeneration, and improved urothelial mucosa to alleviate OHD-induced bladder hyperactivity. PRP could release growth factors to promote angiogenic potential for bladder repair through laminin/integrin-α6 and VEGF/VEGF receptor signaling pathways in the pathogenesis of OHD-induced OAB. Furthermore, PRP enhanced the expression of HA receptors and hyaluronan synthases (HAS), reduced hyaluronidases (HYALs), modulated the fibroblast-myofibroblast transition, and increased angiogenesis and matrix synthesis via the PI3K/AKT/m-TOR pathway, resulting in bladder remodeling and regeneration.

## 1. Introduction

Postmenopausal women with ovary hormone deficiency (OHD) could cause overactive bladder (OAB) symptoms [1,2]. The International Continence Society (ICS) defined OAB syndrome as urinary frequency and nocturia accompanied by urgency with or without urgency urinary incontinence in the absence of urinary tract infection [3,4]. Epidemiological studies have implicated estrogen deficiency as a factor contributing to the increased prevalence of lower urinary tract symptoms (LUTSs) seen with aging in human women [2,5]. Previous reports revealed that approximately 35% of postmenopausal women over 65-year-old suffered from OAB [4]. Although the cause of OAB remains to be found, current medical therapies involving conservative, pharmacological, and surgical treatments are being developed. Clinically, OAB treatment strategies are applied to relieve symptoms. For example, vaginal topical estrogen therapy may help postmenopausal OAB women because it could relieve OAB symptoms, reduce urinary incontinence and improve quality of life [6]. However, postmenopausal women with an estrogen deficiency were usually unwilling to receive estrogen therapy due to worrying hormone-related cancers, endometrial or breast malignancies [7]. In addition to β-AR agonists and antimuscarinic drugs, several therapies, such as phosphodiesterase-5 inhibitors and botulinum toxin injection, were applied to improve OAB symptoms [8]. However, anticholinergic drugs have some side effects, such as dry mouth, constipation, blurry vision, urinary retention, and cognitive effects. Botulinum toxin injection and tibial nerve stimulation for the bladder were the 3rd line treatment owing to invasive procedure with side effects: increasing residual urine, catheter drainage risk, hematuria, nerve pain and injury [8]. There are some risks in surgery for postmenopausal women with OAB, including stress, infection, and urine leakage side effects. It is worthwhile to study the efficacy, safety and cost-effectiveness and to minimize the invasiveness of the treatment for postmenopausal women with OAB.

The previous studies suggested that the ovariectomy (OVX) rat can be used to mimic the postmenopausal state of OHD to induce OAB symptoms [9,10]. The estrogen-deficient rats were found to have a damaged urothelial mucosal layer, a decreased bladder function and an increased prostaglandin level [11]. In OVX rabbits, noticeable vascular degeneration and decreasing vascular density were shown. However, estradiol administration was found to induce angiogenic remodeling characterized by increasing vascular density within detrusor smooth muscle bundles [12,13]. Our previous investigation also revealed that OHD led to diminished bladder compliance, increased interstitial fibrosis and bladder mucosa apoptosis, and strengthened oxidative damage in the OVX rats [14]. Moreover, irritable bladder symptoms become more prevalent at menopause and worsen with increasing vaginal atrophy, including increasing OAB symptoms, recurrent urinary tract infection and non-infective cystitis [15,16]. Bladder trigone was of endodermal origin, and the physical changes correspond with the epidemiological increase in bladder symptoms and cystitis. Moreover, the decline in serum estrogen concentration in postmenopausal women caused the loss of cell proliferation in the vulvovaginal and vesicourethral epithelium, resulting in postmenopausal genitourinary syndrome, including symptoms such as vaginal atrophy. Estrogen therapy has the potential to stimulate the synthesis of hyaluronic acid (HA) via estrogen receptors and entails the activation of the HA synthetase enzyme [17]. As a result, HA has efficacy similar to vaginal estrogens for the treatment of the signs of vaginal atrophy and dyspareunia [18]. Therefore, the OVX-treated rats decreased the expression of cell proliferation as well as angiogenesis-associated protein and altered HA production, which might be attributed to OAB.

According to the previous literature, platelet-rich plasma (PRP) is a platelet-rich centrifugal autologous concentrate. PRP contains not only a high level of platelets but also numerous growth factors, chemokines, cytokines and other proteins. These growth factors of PRP comprised TGF-β, vascular endothelial growth factor (VEGF), epidermal growth factor (EGF), basic fibroblast growth factor (bFGF), platelet-derived growth factor (PDGF) and insulin-like growth factor 1 (IGF-1) [19,20]. Platelets could release cytokines and growth factors stored in alpha granules, among which PDGF and VEGF could promote angiogenesis and provide blood flow nutrients needed for cell proliferation and tissue regeneration; TGF-β and IGF could promote collagen regeneration and fiber strength in damaged tissues [21,22,23,24]. Previous studies mentioned TGF-ß1 may promote cell migration and the synthesis of extracellular matrix (ECM) components such as α-SMA, laminin, collagen and fibronectin [19,20]. PRP could prevent TGF-β1-induced differentiation of fibroblasts to myofibroblasts affected fibrosis by negatively affecting TGF-β1-Smad3-mediated signaling [25,26]. In addition, TGF-1, bFGF, EGF and PDGF up-regulated HA production by increasing hyaluronan synthases (HAS)-mediated HA synthesis in skin fibroblasts [27]. PRP enhanced intracellular TGF-β1 expression, which may regulate HA production [28,29]. Moreover, HA is involved in tissue repair and regeneration, wound healing, and angiogenesis [30,31,32]. For example, intravesical instillation of HA can relieve bladder ulcerative symptoms [33]. Therefore, PRP was taken from animal bodies close to body composition and contained various growth factors to modulate HA production for comprehensive bladder repair.

In the different animal models, PRP intravesical instillation reduced hemorrhage, stimulated angiogenesis, increased activation of mesenchymal stem cells (MSCs) and protected against the tissue fibrosis to improve bladder function and OAB symptoms. In a rat model of ketamine-induced cystitis (KIC), PRP intravesical instillation increased activation of MSCs and protected against the tissue fibrosis [34,35]. In a rabbit model of cyclophosphamide-induced cystitis, intravesical instillation of PRP noticeably reduced hemorrhage, stimulated angiogenesis, strengthened leukocyte infiltration, reinforced mitosis in urothelial mucosa and increased proliferation to improve bladder mucosal repair [36,37]. In addition, PRP application had been reported for modulating tissue regeneration in trauma patients and animals [38]. PRP alone or in combination with bone marrow-derived mesenchymal stromal cells were found to stimulate cell proliferation and differentiation of myogenic progenitors to improve damaged skeletal muscles [39].

In human clinical studies, PRP treatment has a beneficial effect on cell proliferation and collagen production as well as stimulating the production of matrix-degrading enzymes (matrix metalloproteinases) in tenocytes [40], alveolar bone cells [41], osteoblasts [42,43], fibroblasts [43] and bone marrow stem cells [44]. Clinical applications of PRP in the urinary system have been demonstrated for stress urinary incontinence [45], recurrent bacterial cystitis [37], erectile dysfunction and interstitial cystitis/bladder pain syndrome (IC/BPS) [46]. Recent pilot studies have shown that intravesical PRP injections could improve IC symptoms, improve the urothelial regenerative function and reduce chronic inflammation in IC patients. Moreover, PRP injection enhanced urothelial cell proliferation, cytoskeleton remodeling, and barrier protein expression in recurrent urinary tract infections [47]. In clinical trials, repeated intravesical injections of autologous PRP improved bladder pain symptoms and increased bladder capacity in IC/BPS patients who have failed conventional therapies characterized by urgency frequency and suprapubic pain [36,46,48,49]. A pilot study revealed that clinical IC/BPS patients who received repeated PRP intravesical injections once a month for four times showed a meaningful increased urinary IL-2 and IL-8 expression, indicating that PRP may ameliorate IC/BPS symptoms and enhance functional bladder capacity [50,51,52]. IC/BPS could be managed with HA instillation, PRP injection [53], or botulinum toxin A injection [50]. However, the potential effect of PRP therapy in treating OAB symptoms in menopausal women and its underlying mechanism are still unclear.

The effects of growth factors in PRP might promote the differentiation of suburothelial and interstitial fibroblasts to myofibroblasts and increase the expression of HA and ECM remodeling. In view of the importance of HA metabolism in maintaining the integrity of bladder structure, function, and tissue repair, the current study hypothesized that PRP instillation might modulate the expression of HA-metabolizing enzymes and receptors for tissue remodeling, thereby improving bladder repair in OHD-induced OAB. In the current study, the rat model of OVX-induced OAB was applied to explore the therapeutic effect and molecular mechanism of PRP on bladder remodeling and repair via modulating hyaluronan synthesis, including cell proliferation, angiogenesis, myofibroblast differentiation and ECM production. This work intended to develop a non-pharmaceutical PRP therapy directly from autologous blood without the risk of rejection or allergy.

## 2. Results

### 2.1. Serum Estradiol Concentration Was Reduced after OVX Treatment

A bilateral OVX rat model was used to mimic a women’s menopausal status and thereafter induced OAB symptoms. Serum estradiol concentrations were shown in Table 1. After 1 month of OVX treatment, serum estradiol concentrations were significantly reduced in the OVX-treated group, the OVX + PRP group and the OVX + platelet-poor plasma (PPP)-treated group compared with the sham group.

### 2.2. Physical Characteristics

The physical characteristics are shown in Table 1, including water intake, urine output, waist circumference, body weight, bladder weight and the ratio of bladder weight/body weight. After 12 months of bilateral OVX surgery, rats developed a metabolic syndrome as shown in the physical indicators serum biochemical parameters in the OVX group, the OVX + PRP group and the OVX + PPP group. The data revealed there was no significant difference in the amount of water intake and urine output among different groups. In addition, body weight and waist circumference were increased noticeably in groups of OVX, the OVX + PRP group and the OVX + PPP group, as compared to the sham group. Furthermore, bladder weight of the OVX + PRP group and the OVX + PPP group was significantly increased compared to the OVX group. Especially, the ratio of bladder weight and body weight showed significant lower in the OVX group than in the sham group. However, the OVX + PRP group and the OVX + PPP group PRP instillation after OVX showed the recovery of the bladder weight and the ratio of bladder weight/body weight.

The serum parameters associated with the symptoms of metabolic syndrome, including glutamate oxaloacetate transaminase (GOT), glutamate pyruvate transaminase (GPT), triglycerides, cholesterol, low-density lipoprotein (LDL), glucose and lactate dehydrogenase (LDH) (except high-density lipoprotein (HDL) and insulin), were significantly elevated in the OVX group, the OVX + PRP group and the OVX + PPP group, as compared to the sham group. Indeed, PRP and PPP treatments had no significant effect on serum parameters to change lipid profile in the OVX rat model. These results indicated that OHD, in combination with metabolic abnormalities, caused a profound negative effect on the lipid profile. The OVX treated with PRP and PPP groups had limitations in restoring control levels.

### 2.3. PRP Treatment Improved Voiding Dysfunction and Ameliorated Bladder Overactivity

To investigate the pathophysiological relationship between OHD and bladder overactivity, the bladder function was evaluated with urodynamic parameters by cystometrography (CMG) and voiding behavior by metabolic cage. The urodynamic parameters, such as peak micturition pressure, frequency, interval, voided volume and non-voided contraction (star) are shown in Table 1 and Figure 1. The CMG data of the sham group represented stable and regular micturition patterns. The OVX group revealed an increase in micturition frequency and peak micturition pressure compared with the sham group, however, a decrease in micturition interval and voided volume (Table 1 and Figure 1A). In contrast, the OVX + PRP group and the OVX + PPP group showed significantly reduced peak micturition pressure as well as micturition frequency, and increased bladder capacity, as compared with the OVX group (Table 1 and Figure 1A).

Furthermore, tracing analysis of voiding behavior by metabolic cage revealed that the OVX group decreased voiding volume and increased micturition frequency as compared with the sham group (Table 1 and Figure 1B). However, the OVX + PRP group significantly decreased micturition frequency and increased voided volume as compared with those in the OVX group. Therefore, PRP instillation improved voiding dysfunction and ameliorated bladder overactivity. Taken together, the above findings implied that the OVX-treated rats exhibited bladder overactivity with increased micturition frequency and peak micturition pressure, but decreased micturition interval and voided volume. However, treatment with PRP significantly reduced micturition frequency and peak micturition pressure, while increasing bladder capacity. In addition, PRP instillation also improved voiding dysfunction and ameliorated OAB induced by OHD.

### 2.4. PRP Treatment Improved the Bladder Fibrosis

Masson’s trichrome stain was shown to investigate bladder pathologic changes after different treatments (Figure 2). In the sham group (Figure 2A,A’), the urothelial layer (UL; black arrows) had three to five layers, and sparse collagen (blue arrows) was distributed in the suburothelial layer (SL). In contrast, in the OVX group (Figure 2B,B’), exhibited significantly thinner and defective urothelial mucosa in the UL (black arrows) as well as mononuclear cell infiltration (yellow arrows), interstitial fibrosis and collagen accumulation (black arrowheads) in the SL. Similarly, there were mononuclear cell infiltration (yellow arrows) and interstitial fibrosis (black arrowheads) in the SL of the OVX + PRP group and the OVX + PPP group (Figure 2C,C’,D,D’). However, morphological evaluation of the OVX + PRP group (Figure 2C,C’) showed improved OVX-associated bladder damage by increasing the thicker layer of UL and regulating interstitial fibrosis and collagen accumulation (black arrowheads). Particularly, there were many mononuclear cells infiltrated (yellow arrows) and red blood cells (purple arrows) gathered in UL and SL of the OVX + PRP group (Figure 2C,C’). In the OVX + PPP group, bladders exhibited denuded urothelial mucosa (black arrows), mononuclear cell infiltration (yellow arrows), urothelial vocalization (purple arrows) and interstitial fibrosis (blue arrows) (Figure 2D,D’), but such pathological damage was not as profound as those observed in the OVX group.

The distribution of urothelial cell-adhesion marker, E-Cadherin, in the bladder tissue was examined (Figure 2E–H). In the sham group (Figure 2E), the E-Cadherin staining was expressed in intercellular junctions of urothelium, but not in the suburothelium. Conversely, less E-Cadherin staining expression was restricted to the thinner and disrupted urothelium in the OVX group (Figure 2F) compared to the sham group. However, in the OVX + PRP group (Figure 2G) and the OVX + PRP group (Figure 2H), the urothelial staining was enhanced as compared with the OVX group. More importantly, the E-Cadherin immunostaining in the OVX + PRP group was restored to the level of the sham group.

The protein levels of E-cadherin, differentiated marker of urothelial umbrella cells (UPKIII) and inflammation and fibrosis markers (TGF-ß1, fibronectin, collagen I, COX-2 and NFkB-p65) (Figure 2I,J) were evaluated by Western blots. The protein levels of E-Cadherin and UPKIII were significantly decreased in the OVX group as compared with the sham group. However, such proteins were significantly strengthened in the OVX + PRP group and the OVX + PPP group compared with the OVX group. Furthermore, the protein levels of inflammatory and fibrotic markers were shown in Figure 2I,J. In comparison with the sham group, the expression of TGF-ß1 protein was increased by 156.7 ± 29.0% in the OVX group and 128.8 ± 14.5% in the OVX + PRP group versus 91.6 ± 22.0% in the OVX + PPP group. Moreover, the expression of fibronectin was enhanced by 251.3 ± 29.4% in the OVX group and 128.1 ± 20.2% in the OVX + PRP group compared with 259.6 ± 59.6% in the OVX + PPP group. The expression of collagen I was increased by 151.7 ± 35.8% in the OVX group and 104.5 ± 30.4% in the OVX + PRP group versus 158.8 ± 40.4% in the OVX + PPP group. Similarly, the expression of COX-2 protein exhibited a significant increase by 204.6 ± 49.5% in the OVX group and 113.8 ± 35.1% in the OVX + PRP group compared with 103.8 ± 13.2% in the OVX + PPP group. The expression of NFƘB-p65 was increased by 170.7 ± 37.8% in the OVX group and 113.5 ± 29.4% in the OVX + PRP group versus 145.8 ± 33.4% in the OVX + PPP group.

Morphological evaluation and Western blotting analysis revealed that OHD resulting from bilateral OVX led to urothelial atrophy due to reduced expression of urothelial structure markers and exacerbated defects in the urothelial lining, which caused bladder damage. However, weekly instillation of PRP and PPP had a modulatory effect on fibrotic biosynthesis, improved the urothelial barrier, and mitigated bladder injury. These findings suggest that inflammatory and fibrosis markers were significantly reduced in the OVX + PRP group compared to the OVX group. Additionally, PRP may play a crucial role in OHD-induced OAB recovery by inhibiting the expression of NFκB and TGF-β.

### 2.5. Effects of PRP Instillation on Improving OVX-Induced Pathological Alteration

To investigate the therapeutic effect of PRP treatment on the OVX-induced pathological alteration, the expressions of urothelial proliferating markers (Ki67 and CK14) and tight junction markers (Claudin-4 and ZO-1) were analyzed by immunostaining and Western blotting (Figure 3). The Ki67 immunostaining had less distribution in the bladder tissues of the sham group and the OVX group (Figure 3A,B). On the contrary, the Ki67 immunostaining was obviously expressed in the UL and SL in the OVX + PRP group (Figure 3C) and the OVX + PPP group (Figure 3D). Moreover, the co-staining of Claudin-4 (green) and CK14 (red) was widely distributed in the UL of the sham group (Figure 3E). However, the co-staining was restricted to the thin and disrupted urothelium in OVX group (Figure 3F). In comparison with the OVX group, the double labeling of the OVX + PRP group and the OVX + PPP group was markedly expressed in the UL (Figure 3E–H). Particularly, the immunostaining of the OVX + PRP group was stronger than that in the OVX + PPP group. In the OVX + PRP group, the increased proliferation index was assessed by CK14-positive (CK14^+^) and Ki67-positive (Ki67^+^) immunostaining in the UL and SL. These findings revealed that PRP improved urothelial proliferation and tight junction reconstruction and stimulated Ki67^+^ associated with fibroblasts in SL to modulate fibroblast recruitment and improve mucosal regeneration.

Western blotting analysis was performed to further investigate the protein levels of proliferation and urothelial tight junction markers (Figure 3I,J). In comparison with the sham group, the expression of Ki67 protein and CK14 protein were reduced in the OVX group, however, were increased in the OVX + PRP group. Similarly, the expressions of Claudin-4 and ZO-1 protein were decreased in the OVX group compared to the sham group, however, their expressions were enhanced in the OVX + PRP group and the OVX + PPP group compared to the OVX group. Taken together, the increased expression of adhesion protein (E-Cadherin) along with a raise in tight junction protein (Claudin-4 and ZO-1) expression suggested that PRP restored urothelial impermeability. These observations implied that PRP could increase bladder regeneration to promote tissue repair through cell proliferation, differentiation and increasing tight junction reconstruction.

### 2.6. PRP Instillation Modulated Bladder Angiogenic Remodeling and Interstitial Cells to Coordinate Muscle Contractions

To evaluate bladder angiogenic potential and angiogenic remodeling, the angiogenesis markers, including α-SMA, VEGF, VEGF receptor (VEGF-R1 and VEGF-R2), laminin (glycoproteins of ECM) and integrin-α6 (laminin receptor), were analyzed by immunostaining (Figure 4A–H) and Western blotting (Figure 4M and Figure 5N). The myofibroblastic phenotype was evaluated by immunostaining and Western blotting analysis of α-SMA, laminin and vimentin expression. The α-SMA immunostaining (yellow arrows) was widely distributed in myofibroblasts and smooth muscle of microvessels beneath urothelial basal layer, in the SL (lamina propria) and ML of the sham group (Figure 4A), including small artery, small vein, arteriole and venule. In the OVX group (Figure 4B), the α-SMA staining (yellow arrows) was slightly reduced compared to the sham group. However, the expressions were increased in myofibroblasts, microvessels and vessels of the SL and ML in the OVX + PRP group and the OVX + PPP group compared to the OVX group (Figure 4C,D). Particularly, there were many gathered α-SMA positive-myofibroblasts and microvessels beneath urothelial basal layer, lamina propia (yellow arrows) and ML (pink arrows) in the OVX + PRP group (Figure 4C). Similar immunostaining results with α-SMA were obtained for the laminin expression (Figure 4E–H). The staining was significantly increased beneath the urothelial basal layer, in myofibroblasts and microvessels of SL (yellow arrows) and microvessels of ML (pink arrows) in the OVX group (Figure 4F), the OVX + PRP group (Figure 4G) and the OVX + PPP group (Figure 4H) compared to the sham group (Figure 4E).

To elucidate the angiogenic effect of PRP instillation in the pathogenesis of OVX, the levels of angiogenesis associated proteins were quantified by Western blots (Figure 4M,N). In the OVX group, the expressions of these angiogenic proteins were significantly suppressed compared to the sham group, except for laminin and integrin-α6. Moreover, the expressions were noticeably increased in the OVX + PRP group compared to the OVX group (Figure 4M,N). The PRP component included angiogenic factors to promote angiogenic potential through VEGF/VEGF-R signaling pathways for bladder repair in OHD-induced OAB.

Bladder ICs were involved in the pathophysiology of OAB. To elucidate whether the effect of PRP-modulated detrusor muscle contractions to improve bladder capacity in OHD-induced OAB, bladder interstitial cell markers, including vimentin, C-Kit and PDGFR, were performed by immunostaining (Figure 4I–L) and Western blotting (Figure 4O,P). The C-Kit immunostaining (yellow arrows) in the OVX group (Figure 4J) was abundantly distributed in the SL and ML compared to the sham group (Figure 4I). However, the expressions of markers were obviously reduced in the OVX +PRP group (Figure 4K) and the OVX +PPP group (Figure 4L) compared to the OVX group.

The levels of bladder IC markers were quantified by Western blots (Figure 4O,P). In comparison with the sham group, the expression of vimentin was increased by 180.6 ± 30.2% in the OVX group versus 128.2 ± 23.0% in the OVX + PRP group, however, it was reduced by 75.5 ± 16.0% in the OVX + PPP group. Moreover, the expression of C-Kit was enhanced by 168.5 ± 38.0% in the OVX group versus 137.2 ± 27.2% in the OVX + PRP group, however, it was suppressed by 108.0 ± 18.0% in the OVX + PPP group. Similarly, the expression of PDGFR protein was enhanced by 172.5 ± 34.0% in the OVX group, however, it was suppressed by 93.2 ± 29.0% in the OVX + PRP group versus 81.0 ± 28.0% in the OVX + PPP group. The expressions of vimentin, C-Kit and PDGFR markers were noticeably increased in the OVX group compared to the sham group. However, the protein levels were markedly decreased in the OVX + PRP group and the OVX + PPP group compared to the OVX group.

According to the above data, PRP treatment improved angiogenic remodeling by increasing capillary density and promoting ECM for structural scaffolding to coordinate muscle contractions in the pathogenesis of OHD-induced OAB. In addition, OVX-induced abnormal proliferation and differentiation of the interstitial cells (ICs), leading to alternating signal transmission from bladder nerves to smooth muscle cells in the ML during the inflammation process of OVX-induced OAB.

### 2.7. PRP Treatment Triggered Bladder HA Remodeling through HA Receptors

To further explore the effects of PRP on HA receptors, the expression levels of HA receptors (CD44, RHAMM and TLR-4) were quantified by immunostaining and Western blotting (Figure 5). In the sham group (Figure 5A), the E-Cadherin immunoactivity was detected at intercellular junctions throughout the bladder urothelium, including apical, intermediate, and basal layers, but the CD44 immunoactivity was mainly expressed in the basal layer of UL (yellow arrows) and slightly stained in stroma cells (white arrows) of SL (lamina propia). In the OVX group (Figure 5B), the E-Cadherin immunostaining was stained throughout the thin and disrupted urothelial basal layer (yellow arrows); however, the CD44 immunoactivity was mainly expressed in the basal layer of UL (yellow arrows) and stroma cells of SL (white arrows). In the OVX + PRP group (Figure 5C) and the OVX + PPP group (Figure 5D), the CD44 immuno-labeling strongly co-stained with E-Cadherin were strongly distributed in the basal layer of the thickened UL and slightly stained in stroma cells in the SL.

The protein levels of CD44, RHAMM and TLR-4 were performed by Western blotting analysis (Figure 5E,F). In comparison with the sham group, the expressions of CD44, TLR-4 and RHAMM proteins were reduced in the OVX group; however, they were increased in the OVX + PRP group. Therefore, PRP treatment triggered bladder HA remodeling through HA receptors. However, the effect of PRP treatment was more potential than PPP.

The localization of TLR-4 and RHAMM immunoreactivity was assessed (Figure 6). In the sham group (Figure 6A), the TLR-4 immunolabeling co-stained with the E-Cadherin in the UL (yellow arrows), microvessels and vessels (white arrows) of the SL. After OVX treatment (Figure 6B), E-Cadherin and TLR-4 proteins were co-labeled throughout the thin and disrupted urothelium (yellow arrows) and microvessels (white arrows) of the SL compared to the sham group. However, the co-staining levels of the OVX + PRP group (Figure 6C) and the OVX + PPP group (Figure 6D) were strongly distributed in the thickened UL (yellow arrows), including apical, intermediate and basal layers and slightly stained in microvessels and vessels of the SL (white arrows) compared to the OVX group. In some instances, the double-labeling of TLR-4 and E-Cadherin was distributed in hyperplasia urothelium in the OVX + PRP group. Particularly, the level of the OVX + PRP group was much stronger than the OVX + PPP group. Similar results were obtained for RHAMM expression coinciding with TLR-4 (Figure 6E–H).

### 2.8. The PRP Instillation Modulated HA Synthesis and Degradation

Considering that the PRP effect might alter the expression of HA-metabolizing enzymes and receptors for tissue remodeling in a rat model of OHD-induced OAB, the mRNA levels of HA receptors (*CD44*, *RHAMM*, and *TLR-4*), HASs (*HAS 1-3*), hyaluronidases (HYALs) (*HYAL1-4* and *PH20*) in the bladder tissues involving HA synthesis and degradation in the bladder were evaluated by RT-qPCR. The results showed that the mRNA levels of HA receptors (*CD44*, *RHAMM*, and *TLR-4*) (Figure 7A) were significantly declined in the OVX group compared with the sham group, but the levels of these genes were abundantly expressed in the OVX + PRP group and the OVX + PPP group compared with the sham group and the OVX group. In addition, the levels of *HAS2* and *HAS3* (Figure 7B) were significantly decreased in the OVX group compared with the sham group, but the levels were strongly expressed in the OVX + PRP group and the OVX + PPP group compared with the OVX group. In addition, the mRNA expressions of HYALs (*HYAL2*, *HYAL3*, *HYAL4* and *PH20*) were significantly expressed in the OVX group compared with the sham group. However, the expressions of *HYAL2* and *HYAL3* were significantly reduced in the PRP group (Figure 7C). However, there was no significant difference in the *HYAL1* expression among different groups (Figure 7C). Stroma cells, including mesenchymal cells and fibroblast in the SL and interstitium, were the predominant source of HA. The above findings implied that PRP stimulated stromal cells and induced HA receptor expression, enhanced HAS synthesis and inhibited HYAL expression from altering HA metabolism, thus improving bladder remodeling and repair.

### 2.9. Proposed Potential Mechanism of PRP Instillation That Promoted Cell Proliferation and Angiogenesis through PI3K/AKT/m-TOR Pathway, Which Contributed to the Pathogenesis of OHD-Induced OAB

To elucidate whether cellular signaling pathways were involved in regulating bladder remodeling in a rat model of OHD-induced OAB, the level of signaling pathway-related proteins in the bladder, including PI3K (phosphatidylinositol 3-kinase), AKT, m-TOR (mammalian target of rapamycin), e-NOS (endothelial nitric oxide synthase), RAS, ERK1/2 (extracellular signal-regulated kinase 1/2), and transcription factor SOX-9 (stem cell biomarker) was quantified by Western blots (Figure 8). SOX9 is a transcription factor for regulating stem cell proliferation. The expression levels of PI3K, e-NOS and SOX-9 were significantly declined in the OVX group, but the level of AKT, m-TOR, RAS and ERK1/2 were significantly promoted as compared with the sham group. Furthermore, the expressions of all the above proteins were obviously increased in the OVX + PRP group compared to the OVX group, except m-TOR, RAS and ERK1/2. However, there was no significantly different expression of m-TOR, ERK1/2 and SOX-9 between the OVX + PRP group and the OVX + PPP group. Therefore, these observations implied that the OVX + PRP group significantly increased the expression of PI3K, AKT, e-NOS and SOX-9 in the bladder as compared to the OVX group, whereas it reduced the expression of RAS and ERK1/2. The above findings implied that PRP enhanced cell proliferation, modulated fibroblast-myofibroblast transition and increased angiogenesis via PI3K/AKT signaling pathway for bladder repair in OHD-induced OAB.

### 2.10. A Proposed Diagram for the Therapeutic Effect of PRP Improved Bladder Overactivity Induced by OHD in Rat Model

This proposed model established a long-term OHD after bilateral OVX and identified possible mechanisms of detrusor overactivity (Figure 9). PRP instillation could improve OVX-induced damages through modulating HA receptors and/or HA-metabolizing enzymes in a rat model of OHD-induced OAB. Accordingly, the OVX group exacerbated bladder pathological damage and interstitial fibrosis through the NFƘB/COX-2 and the RAS/ERK1/2 signaling pathways. In contrast, PRP instillation for 4 weeks regulated the inflammatory fibrotic biosynthesis, promoted cell proliferation, matrix synthesis and enhanced mucosal regeneration through the HA/PI3K/AKT/SOX-9 signaling pathway to ameliorate OHD-induced bladder dysfunction. Moreover, PRP could promote angiogenic potential through the VEGF/VEGF-R and the PI3K/AKT/e-NOS signaling pathways in the pathogenesis of OHD for bladder repair.

On the other hand, the role of ICs in bladder dysfunction plays an important role in signal transmission from bladder nerves to smooth muscle cells to modulate muscle contractions. However, the effect of PRP or HA on ICs remains unclear. On the basis of the above findings, PRP instillation could recruit stroma cells (fibroblasts and mesenchymal cells), modulate fibroblast-myofibroblast transition (FMT), enhance the expressions of HA receptors and HAS enzymes, reduce HYALs, and increase angiogenesis and the matrix synthesis of SL and ML via the PI3K/AKT/m-TOR pathway, which resulted in bladder regeneration and remodeling. A proposed diagram demonstrated that the effects of PRP could enhance cell proliferation, improve angiogenesis and promote the expression of HA receptors and/or HA-metabolizing enzymes to improve bladder repair in a rat model of OHD-induced OAB.

## 3. Discussion

The above findings revealed that CMG data and voiding behavior indicated significant bladder hyperactivity manifested by an increase in micturition frequency and a decrease in bladder capacity in the OVX-treated rats. Bladder pathological features in the OVX-treated group showed defective urothelial mucosa, mononuclear infiltration, interstitial fibrosis and collagen accumulation. Accordingly, the OVX group exacerbated bladder pathological damage and interstitial fibrosis through the NFƘB/COX-2 and the RAS/ERK1/2 signaling pathways. In contrast, PRP instillation for 4 weeks regulated the inflammatory fibrotic biosynthesis, promoted cell proliferation, matrix synthesis and enhanced mucosal regeneration through the HA/PI3K/AKT/SOX-9 signaling pathway to ameliorate OHD-induced bladder dysfunction. Moreover, PRP could promote angiogenic potential through the VEGF/VEGF-R and the PI3K/AKT/e-NOS signaling pathways in the pathogenesis of OHD for bladder repair. According to the above data, it was suggested that PRP instillation acted on the bladder in two manners. Firstly, PRP instillation ameliorated the inflammatory fibrosis, promoted cell proliferation, enhanced mucosal regeneration and improved urothelial defects to ameliorate OHD-induced bladder dysfunction. Secondly, PRP instillation enhanced the expressions of HA receptors and HAS enzymes, reduced HYALs, modulated fibroblast-myofibroblast transition, and increased angiogenesis as well as ECM synthesis, which resulted in bladder regeneration and repair.

Based on the data of the urodynamic parameter and voiding behavior, both PRP and PPP treatment significantly improved urodynamic parameters compared to the OVX group in Table 1 and Figure 1. According to bladder pathological features by immunostaining and Western blotting, the expression of inflammatory and fibrosis markers [(TGF-ß1, fibronectin, collagen I, COX-2 and NFƘB-p65)] were enhanced in the OVX group in comparison with the sham group; however, the expression was declined in the OVX + PRP group versus the OVX + PPP group. Moreover, the expressions of adhesion protein (E-Cadherin), differentiated urothelial marker (UPKIII), proliferation marker (Ki-67 and CK14), tight junction proteins (Claudin-4 and ZO-1), angiogenesis-related proteins and receptors (α-SMA and VEGF) and HA receptors (CD44 and RHAMM) were reduced in the OVX group compared to the sham group; however, it was increased in the OVX + PRP group versus in the OVX + PPP group. From the above comprehensive results, OHD after bilateral OVX resulted in urothelial atrophy due to less expression of urothelial structure, exacerbated urothelial lining defects and reduced angiogenic remodeling, resulting in bladder damage. Nevertheless, the effects of weekly PRP and PPP instillation modulated the fibrotic biosynthesis, improved urothelial barrier, increased bladder regeneration and angiogenic potential to ameliorate bladder injury in Figure 2, Figure 3 and Figure 4. However, the effect of PRP treatment was more potential than PPP. Although PPP was a plasma fraction that was depleted of platelets, it still contained several cytokines, growth factors and serum proteins.

Kobayashi et al. [54] showed that in PRP preparations, both VEGF and PDGF were significantly more concentrated than PPP. In the scratch assay, PRP was the most effective for wound closure in HUVEC cultures. In the chicken chorioallantoic membrane assay, the α-SMA-staining potency for estimating the number of mature blood vessels was PRP > PPP. The phosphorylation of VEGFR2 by Western blotting in HUVEC cultures was increased in the PRP group compared to the PPP group. This phenomenon could be explained by the direct action of TGF-β, PDGF and VEGF, all of which were concentrated in PRP preparations and capable of stimulating the proliferation of fibroblasts [54]. Additionally, the literature previously published that PRP stimulated VEGF and VEGF receptor to accelerate endothelial cell motility and wound repair. PDGF and VEGF synergistically function facilitated neovascularization during the wound healing process [55]. Additionally, PDGF stimulated the chemotaxis of macrophages and neutrophils and enhanced the secretion of TGF-β from macrophages [56]. We suggested that VEGF, TGF-β, and PDGF provided by PRP in high concentrations cooperatively induced reciprocal interactions between bladder cells and ECM. The effect of PRP improved the neovascularization and therefore increased the blood supply and nutrients influx necessary for cell regeneration in damaged tissue.

Bladder MSCs and fibroblasts have the proliferating ability and migratory capacity, as well as produce abundant ECM components, such as HA, fibronectin, collagens and laminins, which form a loose connective tissue to support urothelial cells [57]. The fibroblast/myofibroblast phenotypes were analyzed by immunostaining and Western blotting of α-SMA, laminin, vimentin and PDGFR expressions. Moreover, the role of ICs in the bladder played a crucial role in the signal transmission from the bladder nerves to the smooth muscle cells and provided pacemaker activity in the smooth muscles. The IC phenotypes were evaluated by immunostaining and Western blotting of vimentin, C-Kit and PDGFR expressions (Figure 4). In addition, macrophages recruited to the inflammatory site and released growth factors, such as TGFβ, PDGF, EGF and FGF, which activated epithelial cells to undergo EMT. In the present study, in the OVX group, the protein expressions of laminin, integrin α6, vimentin, C-Kit, PDGFR and collagen I were enhanced in the OVX group in comparison with the sham group; however, the α-SMA expression was declined (Figure 4). In addition, the expressions of α-SMA, laminin and E-Cadherin were increased in the OVX + PRP group in comparison with the OVX group and the OVX + PPP group, except vimentin, PDGFR and collagen I. On the other hand, bladder ICs were involved in the pathophysiology of OAB. The expressions of IC-related markers (vimentin, C-Kit and PDGFR) were noticeably increased in the OVX group compared to the sham group. However, the protein levels were markedly decreased in the OVX + PRP group and the OVX + PPP group compared to the OVX group. From the above data, chronic inflammatory injury in OVX-induced OAB, fibroblasts or mesenchymal cells and inflammatory cells could accumulate in the bladder interstitium, release inflammatory signals, produce ECM components and then undergo epithelial-mesenchymal transition (EMT). Moreover, these cells express not only mesenchymal markers (α-SMA, collagen I and vimentin) but also epithelial markers (E-Cadherin). OVX-induced abnormal proliferation and differentiation of fibroblasts, mesenchymal cells, inflammatory cells and ICs. These cells might be involved in the pathophysiology of bladder dysfunction. However, PRP instillation could prevent the fibroblast-to-myofibroblast transition (FMT) and inflammatory fibrosis, modulate bladder ICs, improve angiogenic remodeling by increasing capillary density and promoting ECM for structural scaffolding to coordinate muscle contractions in the pathogenesis of OHD-induced OAB.

Stroma cells, including mesenchymal cells and fibroblast, in the SL and interstitium were the predominant source of HA. The HA therapy improved better healing of the urothelium and promoted a more rapid tissue regeneration. HA-mediated cell surface signaling via CD44 was commonly associated with cell migration, cell proliferation and angiogenesis [58,59]. In addition, RHAMM–/– fibroblasts were defective in CD44-mediated ERK1/2 motogenic signaling, leading to defective skin wound repair. Therefore, RHAMM was an essential regulator of CD44-ERK1/2 signaling during wound repair [60]. RHAMM was upregulated in bovine smooth muscle cells after injury, and RHAMM-HA interaction played an important role in the proliferation and migration of smooth muscle cell during wound healing [61]. Two studies by Koyama et al. [62,63] investigated the effect of HAS2 overexpression on angiogenesis and EMT. In our previous study, HA instillations promoted the repair of the damaged GAG layer and enhanced bladder re-epithelialization, proliferation and differentiation in the KIC rat model [33]. On the other hand, the therapeutic effect of PRP instillation decreased inflammatory fibrosis, attenuated oxidative stress, promoted regeneration of urothelial cells, enhanced angiogenesis and neurogenesis, and thereafter restored bladder dysfunction and improved bladder hyperactivity in KIC rats [64]. The present study showed that the immunoactivity of the HA receptors (CD44, RHAMM and TLR-4) was mainly expressed in the basal layer of UL and stroma cells as well as vessels of SL in the OVX group. Additionally, the immunoactivity was strongly distributed in the basal layer of thickened UL and slightly stained in stroma cells in the SL in the OVX + PRP group and the OVX + PPP group compared to the OVX group (Figure 5). Furthermore, the mRNA levels of HA receptors (CD44, RHAMM, and TLR-4) and HASs (HAS2 and HAS3) were significantly declined in the OVX group compared with the sham group, but the levels were abundantly expressed in the OVX + PRP group and the OVX + PPP group compared with the OVX group. In addition, the mRNA expressions of HYALs (HYAL2, HYAL3, HYAL4 and PH20) were significantly expressed in the OVX group compared with the sham group; however, the expressions of HYAL2 and HYAL3 were significantly reduced in the PRP group (Figure 7). The above findings implied that PRP recruited stroma cells (fibroblasts and mesenchymal cells), induced HA receptor expression, enhanced HAS synthesis, and inhibited HYAL expression from altering HA metabolism, thus improving bladder repair. Accordingly, the OVX group exacerbated bladder pathological damage and interstitial fibrosis through the NFƘB/COX-2 and the RAS/ERK1/2 signaling pathways. In contrast, PRP instillation for 4 weeks regulated the inflammatory fibrotic biosynthesis, promoted cell proliferation, matrix synthesis and enhanced mucosal regeneration through the HA/PI3K/AKT/SOX-9 signaling pathway to ameliorate OHD-induced bladder dysfunction. Moreover, PRP could promote angiogenic potential through the VEGF/VEGF-R and the PI3K/AKT/e-NOS signaling pathways in the pathogenesis of OHD for bladder repair.

The phosphoinositide 3-kinase/protein kinase B (PI3K/AKT) signaling pathway was an important signaling pathway of protein synthesis, playing a vital role in cell proliferation, differentiation and apoptosis [65,66]. The PI3K/AKT signaling pathway modulated MSC proliferation in physiological activity, but MSC proliferation was inhibited by adding the specific PI3K inhibitor LY294002 [67]. Overexpression of miR-126 promoted MSCs differentiated into endothelial cells through the activating PI3K/AKT and MAPK/ERK protein phosphorylation. PI3K/AKT-dependent pathway played a protective or regeneration-promoting role after radiation injury [68]. Additionally, SOX9 was a transcription factor for regulating stem cell proliferation. SOX9 could activate the PI3K/AKT signaling pathway and promote cellular proliferation while blocking the PI3K/AKT signaling pathway using inhibitors greatly inhibited the regenerative ability of Sox9-expressing cells [69,70]. Previous results have also shown that cells expressing transcription factor Sox9 were an important factor in the repair and regeneration of radiation-induced lung injury [71]. Furthermore, SOX-9 was involved in regeneration and repair through activation of the PI3K/AKT signaling pathway, suggesting that SOX9 may be a potential clinical therapeutic target in patients with thoracic malignancies. The association of SOX9 expression with PI3K/AKT/m-TOR signaling was validated in clinical samples. However, the present data revealed that the levels of PI3K, e-NOS and SOX-9 were significantly declined in the OVX group, but the level of AKT, m-TOR, RAS and ERK1/2 were significantly promoted as compared with the sham group (Figure 8). In addition, the OVX + PRP group significantly increased the expression of PI3K, AKT, e-NOS and SOX-9 in bladder as compared the OVX group, whereas reduced the expression of RAS and ERK1/2. The above findings implied that the OVX group exacerbated bladder pathological damage and interstitial fibrosis through the RAS/ERK1/2 signaling pathways. In contrast, PRP instillation for 4 weeks promoted cell proliferation, regulated matrix synthesis, and enhanced mucosal regeneration through the HA/PI3K/AKT/SOX-9 signaling pathway to ameliorate OHD-induced bladder dysfunction. Moreover, PRP could promote angiogenic potential through the VEGF/VEGF-R and the PI3K/AKT/e-NOS signaling pathways in the pathogenesis of OHD for bladder repair.

The present work also hoped to develop a non-drug autologous PRP therapy to treat OAB caused by OHD. PRP produced from autologous blood does not cause any allergy or rejection risks and has advantages compared to synthetic materials. Previous studies have shown that at least a five-fold increase in PRP concentration could achieve therapeutic effects [72,73]. However, there is still needed to determine and standardize the ideal concentration, preparation techniques and application protocols to optimize PRP therapeutic effects. In addition, a major limitation of our animal study was the difficulty in obtaining a sufficient volume of unclotted blood from the tail vein of rats. Since 1 mL of blood could be obtained from rats and 0.2 mL of PRP could be generated, additional rats were needed as blood donors. The variabilities in platelet concentration, PRP dosage, equipment and techniques used may alter platelet degranulation characteristics that also could affect the therapeutic effect and outcome in the current study. The optimal concentration of platelets, leukocytes and other plasma components in PRP remains to be clarified.

## 4. Materials and Methods

### 4.1. Animal Model of OVX Rat

The study used female Sprague Dawley rats (Animal Center of BioLASCO, Taipei, Taiwan), weighing 200–250 g and divided them into four groups (10 rats/group): (1) sham group, (2) OVX group: underwent OHD to induce OAB after 12 months of OVX, (3) OVX + PRP group: once weekly PRP for 4 weeks following 12 month of OVX, and (4) OVX + PPP group: once weekly PPP for 4 weeks following 12 month of OVX [74,75]. The OVX-treated rat model was applied to mimic post-menopause in humans to investigate menopause associated with OAB. Under halothane anesthesia, procedures incising the skin about 1 cm in parallel bilateral sides of the rat spine and removing the bilateral ovaries to induce menopause were performed. After OVX operation, the rats were allowed to recover for four weeks and their serum estradiol concentration was checked. Symptoms of OAB, including urinary frequency, urgency, frequency and involuntary detrusor contractions, were analyzed by cystometry, and voiding behavior by metabolic cage was also investigated. The experimental procedures were approved by the Committee for the Use of Experimental Animal of Kaohsiung Medical University (IACUC: KMUH-106086). All experiments were performed in strict accordance with the recommendations in the National Institutes of Health Guide for the Care and Use of Laboratory Animals, and every effort was made to minimize stress/distress in the animals.

### 4.2. Evaluation of Estrogen Hormonal and Biochemical Parameters

Blood samples were collected to assess serum 17β estradiol concentration after bilateral OVX for 1 month. Blood was separated by centrifugation at 4 °C and the serum layer was removed. The micro-titer wells of the 17β estradiol ELISA kit (Cayman Chemical Co., Ann Arbor, MI, USA) were coated with primary antibody for antigenic site of estradiol molecule. The intensity of color development after the addition of substrate solution was inversely proportional to the concentration of estradiol measured by ELISA (Bio-Tek ELX 800, BioTek, Bad Friedrichshall, Germany) [10]. The mean absorbance values of standard and experimental serum samples were calculated for the different groups.

Blood was collected from the heart at the end of the experiment to obtain biochemical data. Serum GOT and GPT activities as well as triglycerides, cholesterol, HDL, LDL, glucose, insulin and LDH concentrations were measured by using an automated analyzer (Selectra Junior Spinlab 100, Vital Scientific, Dieren, Netherlands; Spinreact, Girona, Spain).

Physiological metabolic cage studies for micturition pattern, as previously described [64,76] were performed at the end of the experiment. Rats were placed in individual KDS-TL380 metabolic cages (ADInstruments, Colorado Springs, CO, USA) and maintained for a 24 h familiarization period, and then the micturition frequency and volume of urine output were measured with a cup especially fitted to the MLT0380 transducer (ADInstruments, Colorado Springs, CO, USA). The volumes of 24 h micturition frequency and urine volume were recorded for 3 days. The volumes of water intake and urine output were also measured [77,78]. Their mean values were determined for analyzing micturition patterns.

### 4.3. Cystometrogram (CMG) Study for Bladder Function

To confirm whether OVX-induced voiding dysfunction altered bladder function, the CMGs were performed to record filling pressure, peak micturition pressure, bladder capacity and frequency of non-voiding contractions (without urine leakage during bladder infusion) according to the method previously described [9,77,79]. Under Zoletil-50 (1 mg/kg, intraperitoneal injection; Virbac Laboratories, Carros, France) anesthesia, the rat bladder was emptied, and an indwelling urethral catheter (PE50 tube) was placed for filling the bladder and measuring the bladder pressure before beginning each CMG. Subsequently, the bladders were infused with 0.9% sodium chloride at a steady rate (0.08 mL/min) during which the pressure was measured through the catheter.

A voiding contraction was defined as an increase in bladder pressure that resulted in urine loss. CMG was recorded until the bladder pressure was stabilized and measured at least 5 filling/voiding cycles per rat. Pressure and force signals were amplified (ML866 PowerLab, ADInstrument, Colorado Springs, CO, USA), recorded on a chart recorder, and digitized for computer data collection (Labchart 7 Software, AD Instruments: Windows 7 system, Colorado Springs, CO, USA). Bladder capacity was measured by the amount of saline that infused into the bladder at the commencement of micturition.

### 4.4. PRP and PPP Supernatant Preparation and Intravesical Instillation

5 mL blood was harvested from five healthy donor rats through tail vein under anesthesia. The PRP preparation was carried out by adapting the protocol proposed by Sonnleitner et al. [80,81]. Blood was collected using a 5 mL disposable syringe containing 0.35 mL of 3.8% sodium citrate to 3.15 mL of blood at a ratio of 9:1 per animal and centrifuged at 160× *g* for 20 min at room temperature (22 °C). The whole blood was divided into three layers from top to bottom: plasma layer, platelets, white blood cells (buffy coat) and red blood cells, respectively. The red portion of the lower fraction (red cell component) and the straw-yellow turbid portion of the upper layer (serum component) were shown. The serum component was carefully pipetted and transferred to another new 5 mL vacuum tube and centrifuged at 400× *g* for 15 min at 4 °C to obtain two fractions: the upper fraction on the tube is called PPP and the lower fraction is called PRP.

Compared to PRP, PPP contained smaller quantities of growth factors. Similar amounts of PRP and PPP (0.2 mL) were aspirated and transferred to separate sterile tubes, then activated with 0.1 mL of 10% calcium chloride solution (ScienceLab.com Inc., Houston, TX, EUA). Rat bladder was emptied prior to intravesical instillation of PRP or PPP. 1mL syring and PE-20 tubes were used to inject 0.3 mL of the mixture (0.1 mL of 10% calcium chloride solution and 0.2 mL of PRP or PPP) into the bladder once per week, except for the sham group. Subsequently, the catheters were removed after intravesical instillation (perfusion) of PRP or PPP.

### 4.5. Masson’s Trichrome Staining for Morphological Change

Pathological changes in the bladder were performed using the Masson’s trichrome stain. After fixation with 4% paraformaldehyde for one day at 4 °C, bladder tissue samples were embedded in paraffin, and then 5-µm thin slice sections were prepared for Masson’s trichrome staining (Sigma, Masson’s trichrome Stain Kit HT15, St. Louis, MO, USA) to observe the bladder morphology [77,78,82,83]. The standard Masson’s trichrome staining was used to label connective tissue blue and DSM red. Histology slides stained with Masson’s Trichrome staining were evaluated by two independent pathologists.

### 4.6. Western Blotting Analysis for Protein Expression

Cell lysates were obtained by gently homogenizing of frozen bladder tissue in buffer (50 mM Tris, pH 7.5 and 5% Triton-X100) containing Protease Inhibitor (Pierce, Rockford, IL, USA) and centrifuging (14,000× *g* at 4 °C for 20 min). Equal amounts of total protein (20 μg) were separated into 8% SDS polyacrylamide (SDS-PAGE) gels and transferred to PVDF membranes. After being blocked with non-fat milk (5%), the blotting membranes were then incubated with the primary antibody, including inflammatory and fibrosis markers [transforming growth factor (TGF)-ß1, fibronectin, collagen I, cyclooxygenase (COX)-2 and NFƘB-p65], adhesion protein (E-Cadherin), differentiated urothelial marker (UPKIII), proliferation marker (Ki-67 and CK14), tight junction proteins [Claudin-4 and zonula occludens (ZO)-1], interstitial cell related proteins (vimentin, C-Kit and PDGFR), angiogenesis related proteins and receptors (α-SMA, VEGF, VEGF-R1, VEGF-R2, laminin and integrin-α6), HA receptors (CD44, TLR-4, and RHAMM), and cell signal related proteins(PI3K, m-TOR, RAS, p-AKT, p-ERK1/2, p-P38, SOX-9 and e-NOS). The results obtained were normalized with glyceraldehyde-3-phosphate dehydrogenase (GAPDH; Merck, Kenilworth, NJ, USA, mouse monoclonal IgG, 1:2500, MW: 36 kDa, catalog no. MAB374). The blots were visualized by using enhanced chemiluminescence (ECL) and exposed to Biomax L film (Kodak, Rochester, New York, USA). In each experiment, negative control was shown without primary antibody. All Western blotting analyses were repeated three times and analyzed using Image J version 1.44 software (National Institutes of Health, Bethesda, Rockville, MD, USA). Other materials and procedures used in Western blot experiments are described in Appendix A.

### 4.7. Immunofluorescence Studies for Localization of Protein Expression

According to published methods [64,65,69], the sections of bladder were fixed by 4% paraformaldehyde (overnight, 4 °C), washed with PBS, blocked (10% NGS, 0.5% Triton X-100, in PBS; 1 h, RT), and then incubated (overnight, 4 °C) with the primary antibodies, including E-Cadherin (Proteintech, Chicago, USA, Rabbit polyclonal IgG 1:100), Ki67 (Abcam, Cambridge, UK, rabbit monoclonal IgG 1:100), CK14 (GeneTex, Hsinchu city, Taiwan, rabbit polyclonal IgG, 1:250), Claudin-4 (Invitrogen, Waltham, MA, USA, mouse monoclonal IgG, 1:100), Laminin (Abcam, rabbit polyclonal IgG, 1:100), α-SMA (Abcam, rabbit polyclonal IgG, 1:100), CD44 (Cell Signaling, Danvers, MA, USA, mouse monoclonal IgG1, 1:100), TLR-4 (Proteinech, rabbit polyclonal, 1:100), RHAMM (Proteinech, mouse monoclonal, 1:100) and C-Kit (Bioss, Woburn, MA, USA, rabbit polyclonal IgG, 1:100).

Unbound primary antibodies were washed (0.5% Triton X-100, in PBS; 15 min, RT) and incubated with complementary fluorescence-labeled secondary antibodies (Invitrogen, 1:800) at RT for 1 h. After PBS buffer washing, the nuclei were stained with DAPI and mounted with Prolong Gold anti-fade reagent (Invitrogen). In addition, negative control was performed to illustrate non-specific immunostaining without the primary antibody.

### 4.8. Real-Time Quantitative PCR (RT-qPCR) Analysis

Total RNA extraction from bladder tissues was performed according to the TRIzol manufacturer’s protocol (Invitrogen, Carlsbad, CA, USA). The reaction conditions were set according to the kit instructions. The mRNA was transcribed into cDNA using oligo dT Primers and an RT–PCR Kit (Roche Diagnostics, Mannheim, Germany). SYBR Green I kit (TaKaRa Biotechnology, Kusatsu, Shiga, Japan) was used to detect target mRNA expression in a 7900HT Fast Real-Time PCR system (ABI). Relative quantities of gene transcripts were normalized to ß-actin, and results were analyzed by the 2−ΔΔCt formula. Six samples for each group were shown, and each sample was run in triplicate. The corresponding primers were listed in Table 2.

### 4.9. Statistical Analysis

Analysis of variance, followed by the Bonferroni test and two-way analysis of variance for individual comparison, was conducted for the above experiments. The mean, standard deviation (SD) and *p* values were calculated on triplicate experiments. Student’s *t*-test was used to calculate *p*-values for comparison. *p* < 0.05 was considered statistically significant.

## 5. Conclusions

The therapeutic effect of PRP instillation for 4 weeks regulated the inflammatory fibrotic biosynthesis, promoted cell proliferation, matrix synthesis, increased HA production and enhanced mucosal regeneration through the HA/PI3K/AKT/SOX-9 signaling pathway to ameliorate OHD-induced bladder dysfunction. Moreover, PRP could promote angiogenic potential through the VEGF/VEGF-R and the PI3K/AKT/e-NOS signaling pathways in the pathogenesis of OHD for bladder repair.

## Figures and Tables

**Figure 1 ijms-24-08242-f001:**
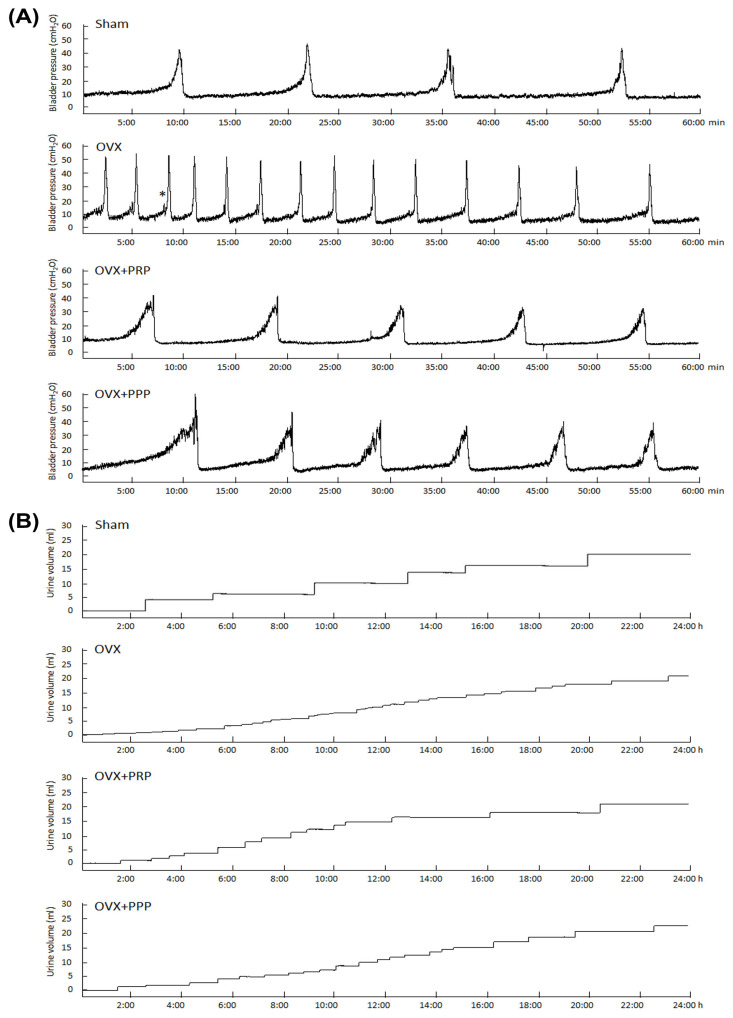
Platelet-rich plasma (PRP) treatment improved voiding behavior and ameliorated bladder overactivity in the ovary hormone deficiency (OHD)-induced overactive bladder (OAB) rat model. Cystometrography parameters (**A**) and tracing analysis of 24 h voiding behavior (**B**) were shown. The OVX group significantly increased micturition frequency, pressure, voiding contractions and non-voiding contractions (asterisk), whereas PRP treatment significantly improved bladder voiding pattern and capacity. *n* = 6 in each group.

**Figure 2 ijms-24-08242-f002:**
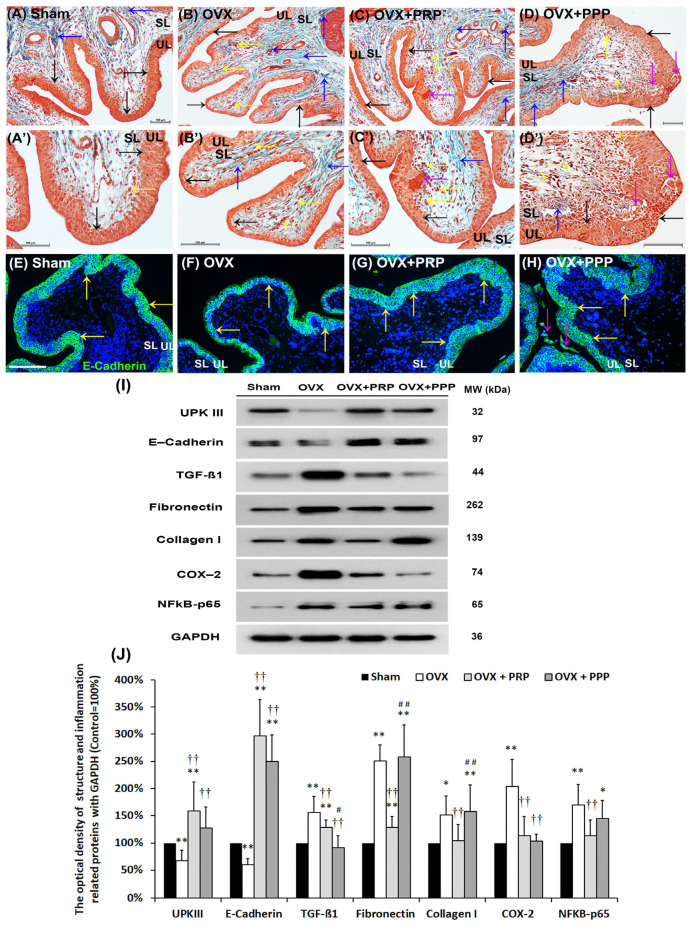
Therapeutic effect of PRP improved OHD-induced pathological alteration by Masson’s trichrome staining, immunostaining and Western blots. (**A**–**D**, **A’**–**D’**): Bladder pathological features of the sham group (**A**,**A’**), the OVX group (**B**,**B’**), the OVX + PRP group (**C**,**C’**) and the OVX + PPP group (**D**,**D’**). Masson’s trichrome stain showed red-stained smooth muscle and green-stained collagen. In the sham group (**A**,**A’**), there were three to five layers of the UL (black arrows), only sparse collagen (blue arrows) and few mononuclear cells (yellow arrows) distributed in the SL (lamina propria). In the OVX group (**B**,**B’**), the morphology of bladder was characterized by a thinner layer (black arrows), much mononuclear cells (yellow arrows), collagen accumulation and increased interstitial fibrosis (blue arrows). In contrast, the OVX + PRP group (**C**,**C’**) and the OVX + PPP group (**D**,**D’**) improved OHD-induced bladder damages by increasing the thicker layer of urothelium (black arrows) and reducing interstitial fibrosis (blue arrows) compared with the OVX group. Additionally, there were many gathered red blood cells (purple arrows) beneath UL and mononuclear cells (yellow arrows) in the SL of the OVX + PRP group. Furthermore, there was a vacuolation in UL (purple arrows) in the OVX + PPP group (D,D’). Scale bar = 100 μm. Original agnification, ×200 (**A**–**D**). Magnification, × 400 (**A’**–**D’**). (**E**–**H**): The distribution of adhesion protein E-Cadherin was expressed by immunostaining. In the sham group (**E**), the E-Cadherin staining was found in intercellular junctions of urothelium. On the contrary, there was less E-Cadherin staining expression in the thin UL of the OVX group (**F**), but the immunostaining in the OVX + PRP group (**G**) and the OVX + PPP group (**H**) was enhanced in UL. In particular, there were some exfoliated epithelial cells in the lumen of the OVX + PPP group. (**I**,**J**): The expressions of urothelial structure (E-Cadherin and UPKIII) and bladder inflammation (TGF-ß1, COX-2 and NFƘB-p65), interstitial fibrosis (fibronectin and collagen I) proteins were analyzed by Western blots. The inflammatory and fibrosis markers were noticeably decreased in the OVX + PRP group and the OVX + PPP group compared to the OVX group. Results were normalized as the sham group (the control group) = 100%. Note: UL, urothelial layer; SL, suburothelial layer; UPKIII, uroplakin III; TGF-ß1, transforming growth factor ß1. Data were expressed as means ± SD for *n* = 6, * *p* < 0.05; ** *p* < 0.01 versus the sham group; ^††^
*p* < 0.01 versus the OVX group; ^#^
*p* < 0.05; ^##^
*p* < 0.01 versus the OVX + PRP group.

**Figure 3 ijms-24-08242-f003:**
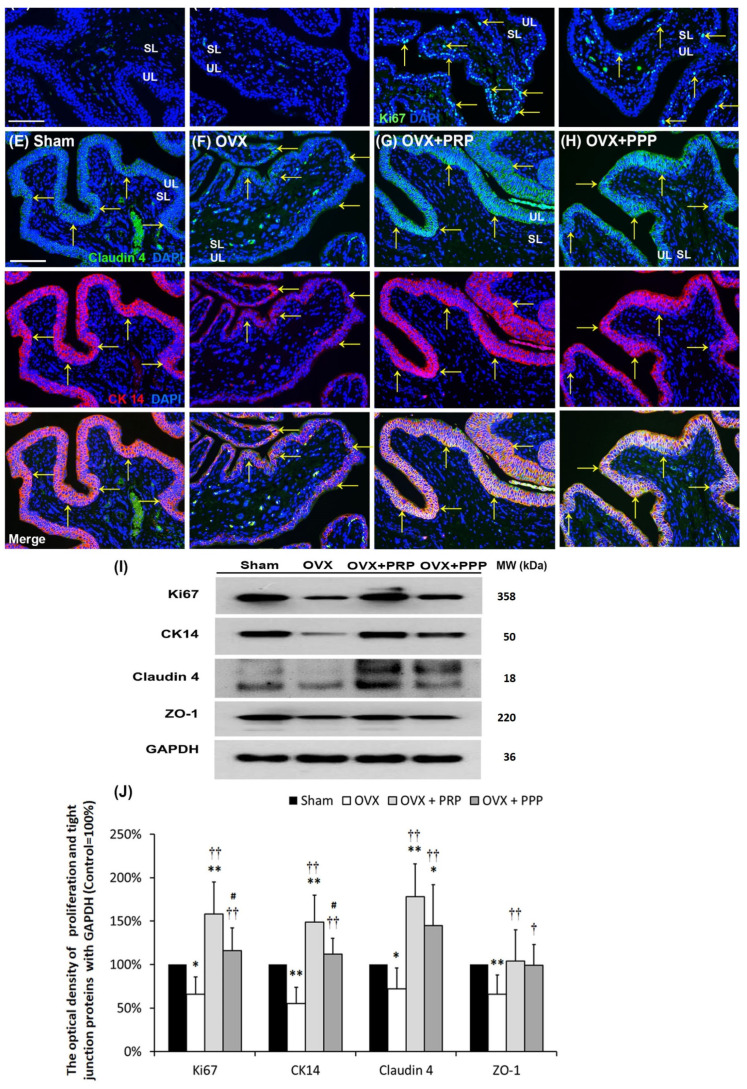
Effects of PRP instillation strengthened proliferation and tight junction reconstruction. The expressions of urothelial proliferating markers (Ki67 and CK14) and tight junction markers (Claudin-4 and ZO-1) were evaluated by immunostaining (**A**–**H**) and Western blotting (**I**,**J**). (**A**–**D**): The Ki67 staining (yellow arrows) had less distribution in the bladder tissues of the sham group (**A**) and the OVX group (**B**). On the other hand, the immunostaining was markedly expressed in the UL and SL in the OVX + PRP group (**C**) and the OVX + PPP group (**D**). The yellow arrows indicated the Ki67-positive cells. (**E**–**H**): Double-labeled analysis of Claudin-4 (fluorescein isothiocyanate; green, **upper panels**) and CK14 (rhodamine; red, **lower panels**) (yellow arrows) was widely distributed in the UL of the sham group (**E**). The OVX group (**F**) showed that double labeling was restricted to the thin and disrupted UL. However, the staining of the OVX + PRP group (**G**) and the OVX + PPP group (**H**) were obviously expressed in the UL compared to the OVX group (**F**). Nuclear DNA was labeled with DAPI (blue). The yellow arrows indicated double staining of Claudin-4 and CK14 in urothelium. (**A**–**D**). (**I**,**J**): Quantifications of the percentage of Ki67, CK14, Claudin-4 and ZO-1 were examined by Western blotting. Western blotting analysis showed that PRP instillation increased the expression of urothelial proliferating markers (Ki67 and CK14) and tight junction markers (Claudin-4 and ZO-1), which improved proliferation and tight junction reconstruction. Note: UL, urothelial layer; SL, suburothelial layer; CK, cytokeratin. Original magnification, × 400 (**A**–**H**). Scale bars = 100 mm. Results were normalized as the sham group = 100%. Data were expressed as means ± SD for *n* = 6, * *p* < 0.05; ** *p* < 0.01 versus the sham group; ^†^
*p* < 0.05; ^††^
*p* < 0.01 versus the OVX group; ^#^
*p* < 0.05 versus the OVX + PRP group.

**Figure 4 ijms-24-08242-f004:**
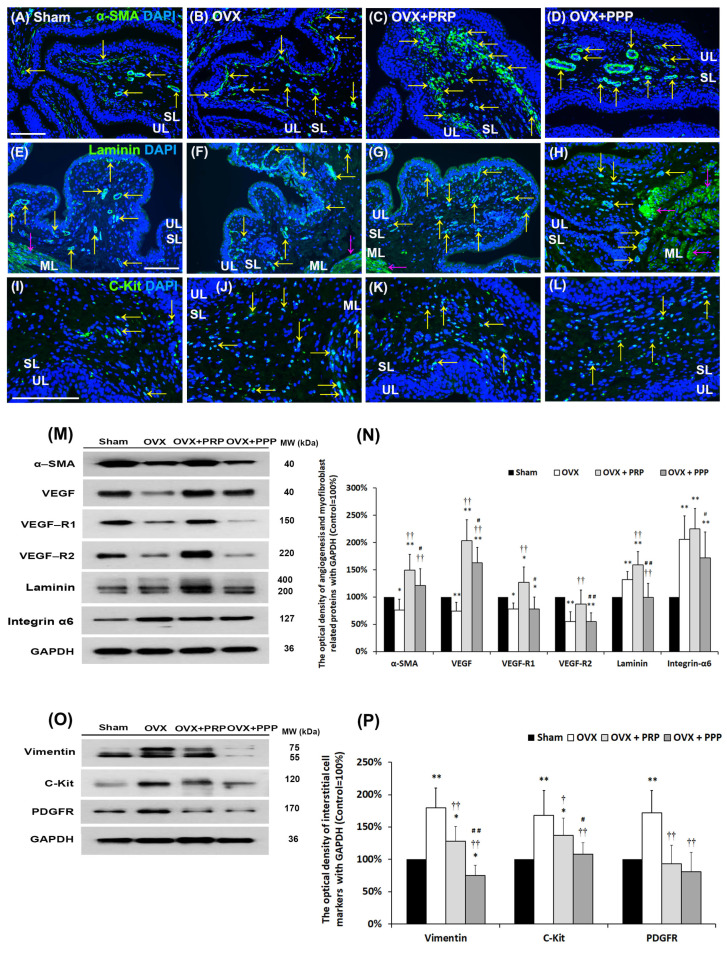
PRP instillation modulated bladder angiogenic remodeling and altered bladder interstitial cells. The angiogenesis related proteins [α-SMA, VEGF, VEGF-R1, VEGF-R2 (VEGF receptor), laminin and integrin-α6 (laminin receptor)] and interstitial cell markers (vimentin, C-Kit and PDGFR) were analyzed by immunostaining (**A**–**L**) and Western blotting (**M**–**P**). (**A**–**D**): In the sham group (**A**), the α-SMA immunostaining (yellow arrows) was widely distributed in the smooth muscle of microvessels beneath urothelial basal layer and vessels in the suburothelial layer (SL) and muscular layer (ML), including small artery, small vein, arteriole and venule. In the OVX group (**B**), the staining was reduced in SL compared to the sham group. However, the staining levels in the OVX + PRP group (**C**) and the OVX + PPP group (**D**) were significantly increased beneath the urothelial basal layer and in the SL (yellow arrows) compared to the OVX group. Particularly, there are many gathered α-SMA positive-myofibroblasts and microvessels beneath urothelial basal layer, lamina propria and ML in the OVX + PRP group. (**E**–**H**): Similar immunostaining results with α-SMA were obtained for the laminin expression. The laminin staining was significantly increased beneath the urothelial basal layer, in myofibroblasts and microvessels of SL (yellow arrows) and microvessels of ML (pink arrows) in the OVX + PRP group (**G**) and the OVX + PPP group (**H**) compared to the sham group (**E**) and the OVX group (**F**). (**I**–**L**): In the sham group (**I**), the c-Kit immunostaining was distributed in bladder SL and ML. The C-Kit immunostaining (yellow arrows) in the OVX group (**J**) was abundantly expressed in the SL and ML compared to the sham group (**I**). However, the expressions of markers were obviously reduced in the OVX +PRP group (**K**) and the OVX +PPP group (**L**) compared to the OVX group. Nucleus was stained by DAPI (blue). Scale bars = 100 mm. (**M**,**N**): Quantifications of the percentage for angiogenesis associated proteins was quantified by Western blots. (**O**,**P**): Quantifications of the percentage for interstitial cell associated proteins were shown. Results were normalized as the sham group = 100%. The protein level in the OVX group was much lower than the sham group. However, the expressions of angiogenesis-associated markers were significantly increased in the OVX + PRP group compared to the OVX group, indicating that PRP improved bladder angiogenic remodeling. Note: UL, urothelial layer; SL, suburothelial layer; ML, muscular layer; α-SMA, alpha-smooth muscle actin; VEGF, vascular endothelial growth factor; PDGFR, platelet-derived growth factor receptors. Data were expressed as means ± SD for *n* = 6, * *p* < 0.05; ** *p* < 0.01 versus the sham group; ^†^
*p* < 0.05; ^††^
*p* < 0.01 versus the OVX group; ^#^
*p* < 0.05; ^##^
*p* < 0.01 versus the OVX + PRP group.

**Figure 5 ijms-24-08242-f005:**
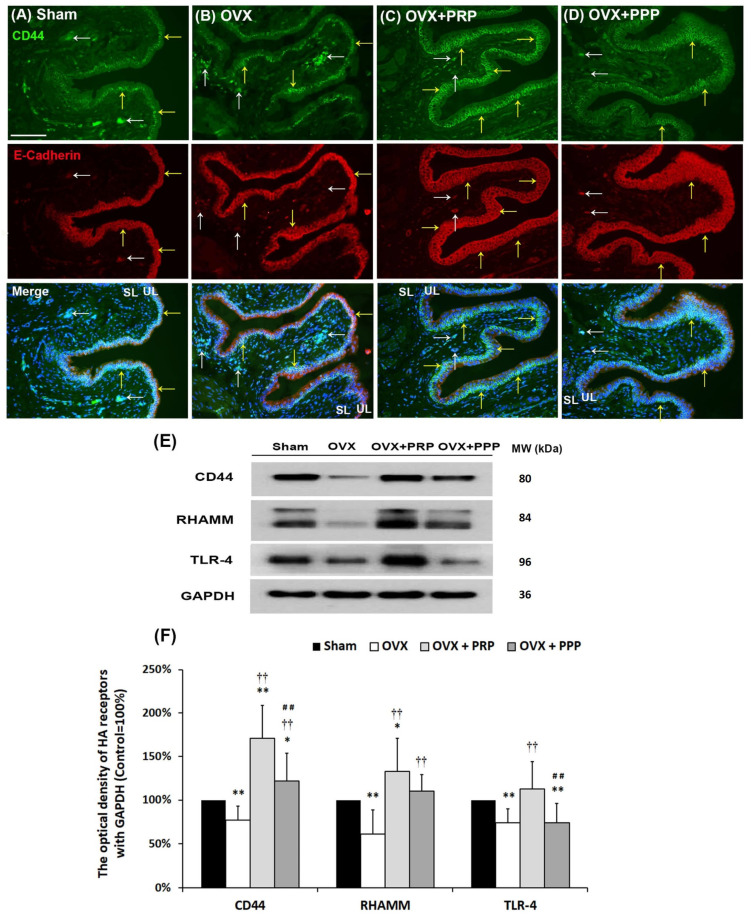
Effects of PRP instillation enhanced the HA receptor expression. The expressions of HA receptors in the bladder were analyzed by immunostaining (**A**–**D**) and Western blotting (**E**,**F**). (**A**–**D**): In the sham group (**A**), CD44 expression (green) co-stained with E-Cadherin (red) was mainly distributed in the basal layer of UL (yellow arrows), and slightly co-labeled in stroma cells and vessels of SL (white arrows). (**B**): In the OVX-treated group, CD44 protein was mainly expressed throughout the ulcerated and disrupted urothelium (yellow arrows) and the stromal cells as well as vessels (white arrows) of the SL. (**C**,**D**): In the OVX + PRP group and the OVX + PPP group, CD44 positive cells were predominantly expressed in the basal layer of hyperplasia urothelium (yellow arrows) and were slightly stained in vessels of the SL (white arrows). (**E**,**F**): The protein levels of HA receptors (CD44, RHAMM and TLR-4) were investigated in the different groups by Western blotting analysis and normalized to the sham group. The yellow arrows indicated double staining of RHAMM and E-Cadherin in urothelium, and white arrows indicated double staining in stroma cells. Note: UL, urothelial layer; SL, suburothelial layer; HA, hyaluronic acid; RHAMM, receptor for HA-mediated motility; TLR-4, Toll-like receptor-4; GAPDH, glyceraldehyde-3-phosphate dehydrogenase. Data were expressed as means ± SD for *n* = 6, * *p* < 0.05; ** *p* < 0.01 versus the sham group; ^††^
*p* < 0.01 versus the OVX group; ^##^
*p* < 0.01 versus the OVX + PRP group.

**Figure 6 ijms-24-08242-f006:**
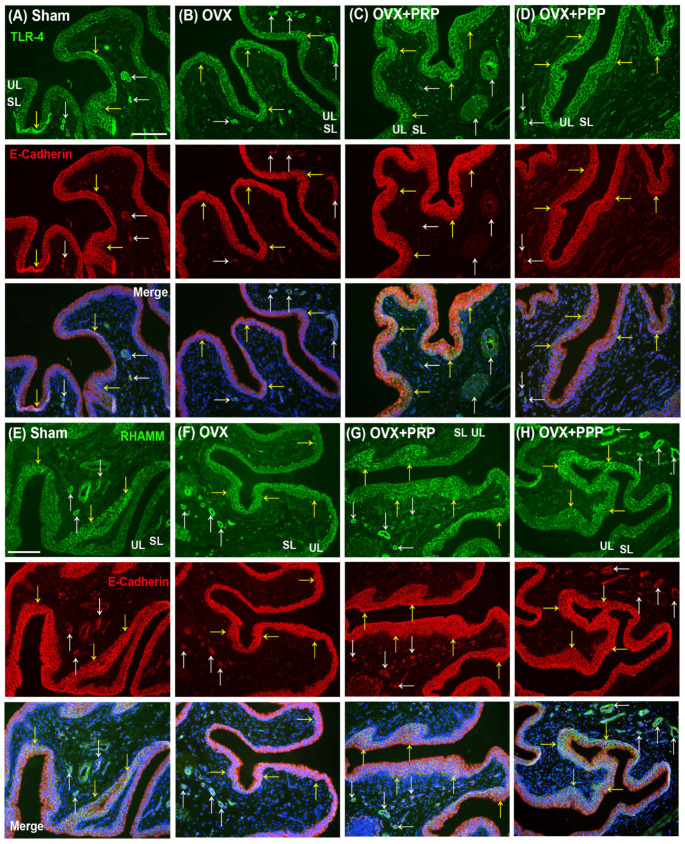
Immunofluorescence analysis was conducted to assess the expression of HA receptors (TLR-4 and RHAMM) in the bladder following treatments. (**A**–**D**): The double-labeling of TLR-4 (green) and E-Cadherin (red) was performed in the bladder. In the sham group (**A**), the double-labeling was widely distributed in the urothelial basal layer (yellow arrows) as well as microvessels and vessels (white arrows) in the SL and ML. In the OVX group (**B**), the co-staining of UL and SL was reduced compared to the sham group. However, the co-staining levels of the OVX + PRP group (**C**) and the OVX + PPP group (**D**) were increased throughout the bladder urothelium, including apical, intermediate and basal layers, compared to the OVX group. Particularly, the level of the OVX + PRP group in the SL and ML was much stronger than the OVX + PPP group. (**E**–**H**): The double-labeling of RHAMM (green) and E-Cadherin (red) was shown in the bladder of the different groups. Similar results were obtained for RHAMM expression coinciding with TLR-4. Yellow arrows indicated the double staining of RHAMM and E-Cadherin in urothelium, while the white arrows indicated the double staining in stroma cells and vessels. Note: UL, urothelial layer; SL, suburothelial layer; ML, muscular layer; RHAMM, receptor for HA-mediated motility; TLR-4, Toll-like receptor-4. Scale bars = 100 mm. Original magnification, ×400 (**A**–**H**).

**Figure 7 ijms-24-08242-f007:**
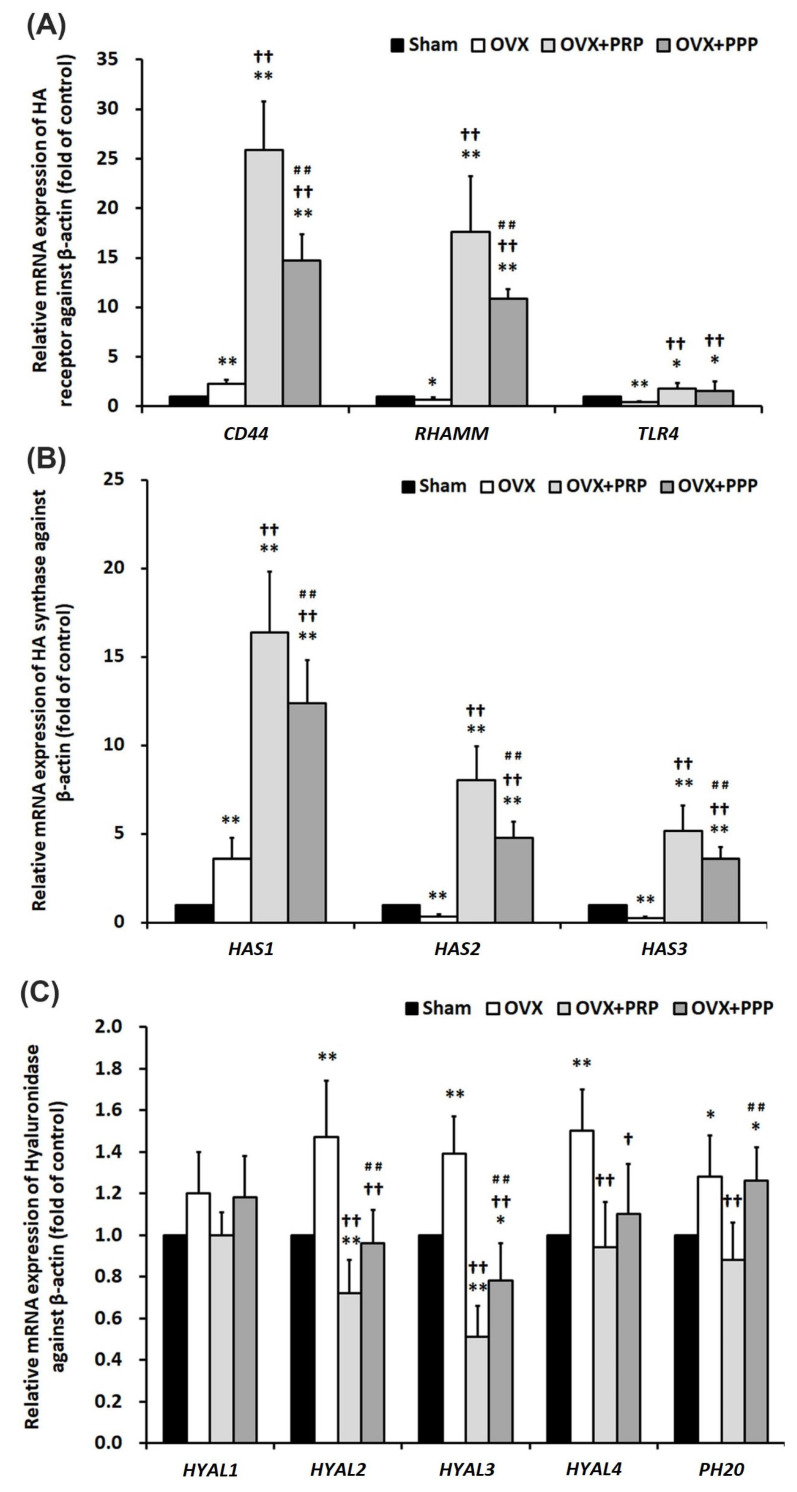
Real-time quantitative PCR of the mRNA levels of HA receptors (*CD44, TLR-4* and *RHAMM*); HA synthase (*HAS1* to *HAS3*) and hyaluronidase [*HYAL1* to *HYAL4* and sperm adhesion molecule (*PH20*)] involved in HA synthesis and degradation of bladder tissue. (**A**) The mRNA levels of HA receptors (*CD44*, *RHAMM*, and *TLR-4*), and (**B**) HA synthase (*HAS2* and *HAS3*) were significantly declined in the OVX group compared with the sham group, but the levels were abundantly expressed in the OVX + PRP group and the OVX + PPP group compared with the sham group and the OVX group. In addition, the mRNA expressions of HYALs (*HYAL2*, *HYAL3*, *HYAL4* and *PH20*) were significantly expressed in the OVX group compared with the sham group. Moreover, the expressions of *HYAL2* and *HYAL3* were significantly reduced in the PRP group compared to the OVX group (**C**). Data were expressed as means ± SD. *n* = 6 in each group (**A**–**C**). Note: *RHAMM*, receptor for HA-mediated motility; *TLR-4*, Toll-like receptor-4; *HYAL*, hyaluronidase; *PH20*, sperm adhesion molecule. * *p* < 0.05; ** *p* < 0.01 versus the sham group; ^†^
*p* < 0.05; ^††^
*p* < 0.01 versus the OVX group; ^##^
*p* < 0.01 versus the OVX + PRP group.

**Figure 8 ijms-24-08242-f008:**
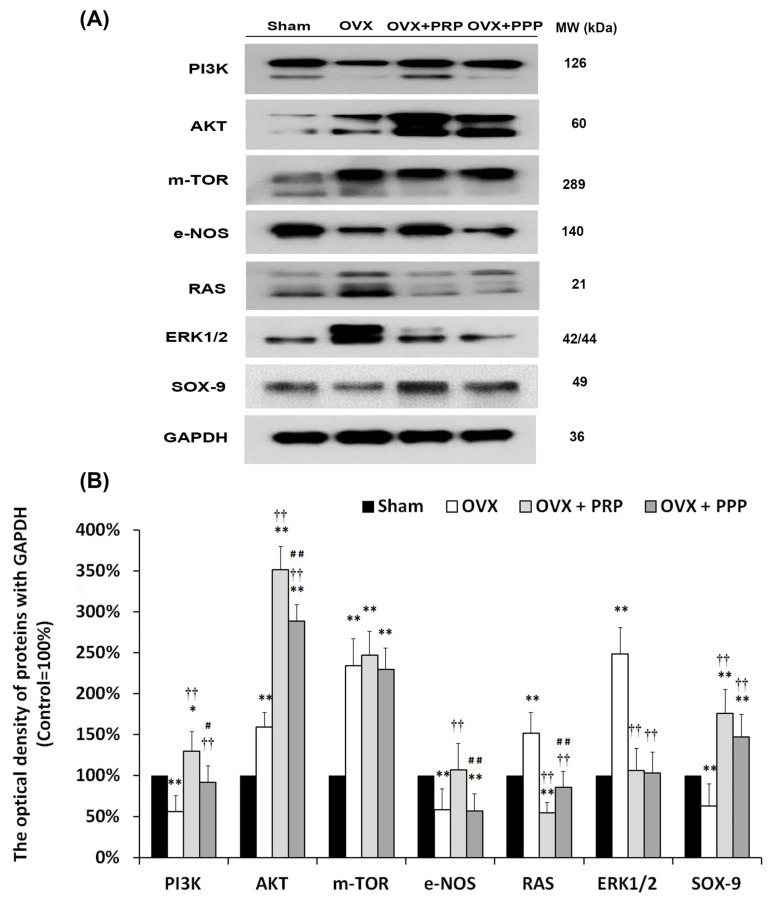
Proposed potential mechanism of PRP instillation that promoted cell proliferation and angiogenesis through PI3K/AKT/m-TOR pathway contribution to the pathogenesis of OHD-induced OAB. (**A**,**B**): The levels of signaling related proteins in the bladder, including PI3K, AKT, m-TOR, e-NOS, RAS, ERK1/2, and SOX-9, were quantified by Western blots. In the OVX group, the expressions of PI3K, e-NOS and SOX-9 proteins were significantly declined, but the levels of AKT, m-TOR, RAS and ERK1/2 were significantly promoted as compared with the sham group. Additionally, the expressions of all the above proteins were obviously increased in the OVX + PRP group (except m-TOR, RAS and ERK1/2) as compared to the OVX group. Note: e-NOS, endothelial nitric oxide synthase; ERK, extracellular signal-regulated kinase; GAPDH, glyceraldehyde-3-phosphate dehydrogenase; m-TOR, mammalian target of rapamycin; PI3K, phosphatidylinositol 3-kinase; SOX-9, SRY-box- 9. Data were expressed as means ± SD for *n* = 6, * *p* < 0.05; ** *p* < 0.01 versus the sham group; ^††^
*p* < 0.01 versus the OVX group; ^#^
*p* < 0.05; ^##^
*p* < 0.01 versus the OVX + PRP group.

**Figure 9 ijms-24-08242-f009:**
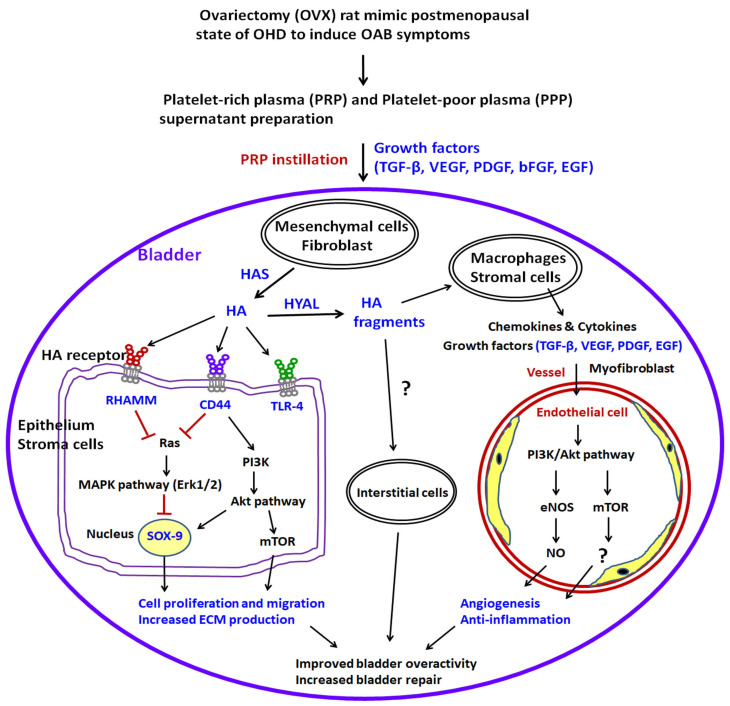
A proposed diagram for the therapeutic effect of PRP improved bladder overactivity induced by OHD in rat model. Accordingly, the OVX group exacerbated bladder pathological damage and interstitial fibrosis through the NFƘB/COX-2 and the RAS/ERK1/2 signaling pathways. In contrast, PRP instillation for 4 weeks regulated the inflammatory fibrotic biosynthesis, promoted cell proliferation, matrix synthesis and enhanced mucosal regeneration through the HA/PI3K/AKT/SOX-9 signaling pathway to ameliorate OHD-induced bladder dysfunction. Moreover, PRP could promote angiogenic potential through the VEGF/VEGF-R and the PI3K/AKT/e-NOS signaling pathways in the pathogenesis of OHD for bladder repair. Note: bFGF, basic fibroblast growth factor; CD44, cluster of differentiation 44; EGF, epidermal growth factor; EMT, epithelial-mesenchymal transition; ERK, extracellular signal-regulated kinase; HA, hyaluronan; HAS, HA synthases (HAS 1-3); HYAL, hyaluronidases (HYAL1-4 and PH20); MAPK, mitogen-activated protein kinase; m-TOR, mammalian target of rapamycin; NO, Nitric oxide; OAB, overactive bladder; PDGF, platelet-derived growth factor; PI3K, phosphatidylinositol 3-kinase; RHAMM, receptor for HA-mediated motility; TLR-4, Toll-like receptor-4; TGF-β, transforming growth factor-β; VEGF, vascular endothelial growth factor.

**Table 1 ijms-24-08242-t001:** Physical characteristics, serum and urodynamic parameters for different experimental groups.

Variable	Sham	OVX	OVX + PRP	OVX + PPP
No. rats	10	10	10	10
Serum estradiol conc. (pg/mL) before treatment	31.6 ± 4.2	32.6 ± 3.4	31.8 ± 2.9	33.0 ± 2.5
Serum estradiol conc. (pg/mL) after treatment	32.8 ± 3.6	16.2 ± 3.5 **	15.8 ± 2.1 **	16.9 ± 2.3 **
Physical characteristics				
Water intake (mL/24 h)	38.7 ± 8.9	35.7 ± 7.7	37.7 ± 5.3	36.0± 6.1
Urine output (mL/24 h)	22.6 ± 5.2	17.0 ± 4.5	18.6 ± 3.0	17.9 ± 3.3
Waist circumference (cm)	17.6 ± 2.6	22.6 ± 3.8 *	21.4 ± 2.5 *	21.8 ± 3.0 *
Body weight (g)	345.2 ± 28.6	469. 8 ± 41.4 **	450.7 ± 37.6 **	459.3 ± 38.2 **
Bladder weight (mg)	218.0 ± 16.6	196.7 ± 25.8 *	230.0 ± 19.7 ^†^	226.7 ± 16.8 ^†^
The ratio of bladder weight (mg)/ body weight (g)	0.60 ± 0.09	0.41 ± 0.06 **	0.55 ± 0.07 ^†^	0.49 ± 0.05 *^,†^
Serum parameters				
GOT (U/dL)	42.3 ± 33.1	126.3 ± 38.0 *	116.3 ± 23.1 *	121.8 ± 18.2 *
GPT (U/dL)	38.0 ± 6.0	79.0 ± 12.6 *	64.7 ± 12.3 *	69.7 ± 17.9 *
Triglyceride (mg/dL)	76.7 ± 13.3	128.8 ± 27.8 **	83.3 ± 14.2 **	89.8 ± 33.3 **
Cholesterol (mg/dL)	66.9 ± 14.8	162.8 ± 12.2 **	150.7 ± 9.5 **	163.3 ± 8.9 **
HDL (mg/dL)	68.9 ± 7.5	61.3 ± 10.0	54.7 ± 7.8	53.3 ± 8.7
LDL (mg/dL)	13.5 ± 0.7	50.7 ± 4.5 **	40.9 ± 6.5 **	46.7 ± 1.8 **
Glucose (mg/dL)	102.5 ± 10.7	127.8 ± 18.6 *	117.0 ± 7.7	118.3 ± 11.0
Insulin (Bayer) (mU/L)	0.4 ± 0.05	0.5 ± 0.06	0.4 ± 0.04	0.5 ± 0.06
LDH	133.8 ± 12.8	394.5 ± 41.0 **	254.2 ± 27.9 **^,††^	274.0 ± 34.4 **^,††^
Urodynamic parameters				
Frequency (No. voids/1 h)	4.2 ± 0.8	10.8 ± 2.2 **	4.5 ± 1.0 ^††^	5.8 ± 1.2 *^,††^
Peak micturition pressure (cmH_2_O)	35.6 ± 5.0	51.6 ± 6.2 *	37.9 ± 3.1	41.8 ± 4.5 ^†^
Voided volume (mL)	2.1 ± 0.4	0.7 ± 0.3 **	1.6 ± 0.4 ^†^	1.3 ± 0.3 *^,†^
No. non-voiding contractions between micturition (No./60 mins)	0	1.3 *	0	0

Footnote: OVX, surgical ovariectomy; PRP, platelet rich plasma; PPP, platelet-poor plasma; GOT, glutamate oxaloacetate transaminase; GPT, glutamate pyruvate transaminase; LDL, low-density lipoprotein; HDL, high-density lipoprotein; LDH, Lactic dehydrogenase; Value is mean ± SD. * *p* < 0.05; ** *p* < 0.01 versus the sham group. ^†^
*p* < 0.05; ^††^
*p* < 0.01 versus the OVX group.

**Table 2 ijms-24-08242-t002:** Primer sequences for quantitative reverse transcription PCR (*RT-qPCR*).

Gene Name	Accession Number	Forward Primer Sequence (5’→3’)	Reverse Primer Sequence (5’→3’)	Tm (°C)	Product Size (bp)
*β* *-actin*	NM_007393	ATCTCCTTCTGCATCCTGTCGGCAAT	CATGGAGTCCTGGCATCCACGAAAC	59	145
Hyaluronan (HA) receptor					
*CD44*	NM_012924.2	AGAAGGTGTGGGCAGAAGAA	AAATGCACCATTTCCTGAGA	59	116
*TLR4*	NM_019178.1	GGGTGAGAAACGAGCT	TTGTCCTCCCACTCGA	59	101
*RHAMM*	NM_012964.2	TGCAAAGCCAGTCACTTCTG	GACATTCCTCTCGGAGGTCA	59	101
Hyaluronan synthase (HAS)					
*HAS1*	NM_172323.1	AGTATACCTCGCGCTCCAGA	ACCACAGGGCGTTGTATAGC	59	120
*HAS2*	NM_013153.1	ATAAGCGGTCCTCTGGGAAT	CCCTGTTGGTAAGGTGCCTA	59	124
*HAS3*	NM_172319.1	AGCAGCGTGAGGTACTGGTA	AGTCCTCCAGGAACTGCTGA	60	130
*PH20*	NM_053967.2	ACTATCCTCACATAGATGCACAGC	TCGACTCGACTTCAAATCTTTCTT	60	524
Hyaluronidases (HYAL)					
*HYAL1*	NM_207616.1	ATGACCAGCTAGGGTGGTTG	CTCTTGCACACGGTATCGAA	59	119
*HYAL2*	NM_172040.2	AGGCCTGTATCCACGTTTTG	GTTCCACAGCTTCCTTCAGC	59	107
*HYAL3*	NM_207599.2	GTGTTCGAGCTGTGGTGTGG	GGGGATCTTCCTCCAAGACC	59	122
*HYAL4*	NM_001100780.1	ACCCATCAATGGTGGTCTTC	GCGCCAATATTCCCAGTCTA	59	133

Footnote: *TLR4*, Toll-like receptor 4; *RHAMN*, receptor of Hyaluronan mediated motility; *PH20*, sperm adhesion molecule 1.

## Data Availability

Not applicable.

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
