# Peer review of "Effects of Therapeutic Platelet-Rich Plasma on Overactive Bladder via Modulating Hyaluronan Synthesis in Ovariectomized Rat"

_ijms, 2023, doi:10.3390/ijms24098242_

Round 1

Reviewer 1 Report

This is an interesting manuscript on a very important topic in the field of bladder dysfunction and lower urinary tract symptoms. The research design deserves merit, especially because it employed a number of different approaches using feasible model systems. However, there are numerous issues that, if addressed, may significantly improve the quality of the manuscript.

Comments:

- Page 1, lines 4 and 5: The names of authors end with “and *,†”. Name of the last author may be missing.

- The abstract is concise presenting a nice summary of the article.

Introduction:

- The introduction is somewhat diffuse. The introduction should provide a better focused and clear background pertaining to the main scope of the study.

- The rational for using PRP against OHD-induced OAB should be described clearly. It should be clarified that PRP therapy is aimed at alleviating bladder symptoms caused by OHD while having no effect on OHD itself that is the main cause of bladder dysfunction in the OVX model.

-  There are several important recent articles pertaining to the clinical application of PRP for bladder dysfunction that could be cited in the introduction section. The literature also suggests that PRP therapy may reduce bladder symptoms temporarily as it acts locally on the bladder, but PRP doesn’t cure the main cause of bladder dysfunction. Therefore, structural and functional changes of the bladder may reappear over time after PRP therapy is stopped.

Results:

- Data presentation is the weakest part of this article. The data, as presented, is somewhat unclear and difficult. Unnecessary details are provided while crucial findings are not described clearly and remain obscure. The data, as presented, suggest that PRP can reverse almost all structural and functional changes of the bladder induced by OHD in the OVX model, which is somewhat surprising.  

- As stated earlier, OHD is the main cause of bladder dysfunction in the OVX model and PRP has no significant effect on OHD. Thus, PRP may somewhat improve OHD-induced bladder symptoms but may not be capable of normalizing all functional, structural, and molecular changes in the OVX model to the control levels. Please reanalyze the data or provide clarifications.  

- The data in table 1 should be described clearly. Please provide a legend for table 1 and provide a clear description of the findings. Under urodynamic data, please clarify which data is from metabolic cage studies and which data is from CMG?

- What was the reason for measuring lipid profile (cholesterol, triglycerides, LDL, HDL)? What is the link between OHD, PRP therapy, and lipid profile? Did PRP therapy significantly change lipid profile? If so, why?

- It seems that PPP significantly diminished voiding frequency, voided volume, peak micturition pressure, and number of non-voiding contractions. The data in table 1 suggest that PRP and PPP had similar effects on the urodynamic parameters. Please provide clarifications and describe how PPP produced significant effects through which downstream mechanisms.

- Table 1 shows that PPP improved voided volume but the CMG data in figure 1 shows the opposite. Did PPP improve both voided volume and CMG changes? Please provide clarifications and describe how PPP acted on the dysfunctional bladder and improved urodynamic changes.

- Figure 1 suggests that PRP therapy completely normalized bladder pressure, frequency of contractions, and voided volume of the OVX group to the levels recorded in sham controls. The effects of PRP shown in figure 1 are dramatic and somewhat surprising. Did PRP cure all functional changes in the OVX group and normalized them to the control levels? Downstream mechanisms mediating such a dramatic effect of PRP should be described clearly.

- Structural data also suggest that PRP therapy reversed almost all structural changes of the OVX group to the control levels. However, the data as presented is not clear and need clarifications.

- The trichrome, immunofluorescence, and western blotting data in figure 2 do not show the findings clearly. The data is poorly presented. The descriptions provided in figure legend cannot be found in the corresponding sections of figure 2.

- Same issue with figure 3. Unclear and poorly defined figures are presented. The figures do not show the findings clearly. Detailed descriptions made in figure legend cannot be found in the figures.  

- Sections A-L of figure 4 do not show the findings clearly. No distinction is made between arterioles, venules, and lymphatic ducts.  It seems that all those structures are interpreted as arterioles. Sections M and O are blurry and unclear.

- The immunofluorescence data in figure 6 do not show the findings clearly. Detailed descriptions made in figure 6 legend cannot be found in the figures. 

- Sections A, B, C, and D of figure 7 do not show the data clearly. It is not clear what the authors intend to show in the fluorescence section of this figure.

- Figure 8A is blurry and unclear.

- Sections B, C D and E of figure 9 are not needed and should be removed.

Discussion:

- The authors should provide a comprehensive and analytical discussion of the findings and clarify how PRP therapy produced such dramatic effects and completely normalized all structural and functional changes of the OVX group to the levels of sham control group while the main cause of bladder dysfunction (OHD) remained untreated

- The discussion lacks cohesion as to how the findings presented in this paper coalesce into new information pertaining to PRP therapy and bladder dysfunction.

- Some paragraphs of the discussion are more appropriate for the introduction, wherein it justifies the experiments performed but does not place the new findings in context of what is already known about PRP therapy for bladder dysfunction.

- A deeper discussion that places the new findings in a broader context whilst offering alternative interpretations of the data may enhance the quality of this article.

Materials and Methods:

The materials and method section is well done.

- Figure 10 in the method section is not needed and should be removed.

Author Response

Reviewer comments:

Reviewer #1:

The paper submitted is a very interesting study investigating the pathophysiological mechanism of bladder dysfunction in ketamine induced ulcerative cystitis in a rat model and the therapeutic anti-inflammatory and regenerative effects of intravescical instillation of platelet-rich plasma on tissue remodeling. The study was performed and described very well, however there are some areas that could be improved. There are some minor inaccuracies in the manuscript text. Literature references should be extended. Specific points are listed below.

This is an interesting manuscript on a very important topic in the field of bladder dysfunction and lower urinary tract symptoms. The research design deserves merit, especially because it employed a number of different approaches using feasible model systems. However, there are numerous issues that, if addressed, may significantly improve the quality of the manuscript.

Comments:

  • Page 1, lines 4 and 5: The names of authors end with “and *,†”. Name of the last author may be missing.

Response: Thanks for your professional recommendation. As suggested by the reviewer, the following information has been revised to the author section on page 1

, as “ * Corresponding author: juanuro@gmail.com (Yung-Shun Juan); ed100464@edah.org.tw (Hung-Yu Lin)

† These authors contributed equally to this work.

  • The abstract is concise presenting a nice summary of the article.

Response: Thank you for the compliment.

  • Introduction: The introduction is somewhat diffuse. The introduction should provide a better focused and clear background pertaining to the main scope of the study.

Response: As suggested by the reviewer, we have revised and reorganized the background and the main scope in the Introduction section on page 2 to 4.

  • Introduction: The rational for using PRP against OHD-induced OAB should be described clearly. It should be clarified that PRP therapy is aimed at alleviating bladder symptoms caused by OHD while having no effect on OHD itself that is the main cause of bladder dysfunction in the OVX model.

Response: As suggested by the reviewer, we have added the rational for using PRP against OHD-induced OAB in the Introduction section on page 3 and 4, which should provide a clearer understanding of the study's goals and methods. We also clarified that PRP therapy is aimed at alleviating bladder symptoms caused by OHD while having no effect on OHD itself. Additionally, in Table 1, the physical characteristics and serum parameters of the OVX treated with PRP and PPP groups have been shown to have limitations in restoring control levels. This information is important for understanding the efficacy of PRP treatment in the context of the study. Based on the present data, PRP instillation for four weeks has been shown to regulate inflammatory fibrotic biosynthesis, promote cell proliferation and matrix synthesis of stroma, enhance mucosal regeneration, and improve urothelial mucosa. These effects suggest that PRP treatment can help alleviate OHD-induced bladder hyperactivity, which is a promising result for the potential clinical application of PRP therapy

  • Introduction: There are several important recent articles pertaining to the clinical application of PRP for bladder dysfunction that could be cited in the introduction section. The literature also suggests that PRP therapy may reduce bladder symptoms temporarily as it acts locally on the bladder, but PRP doesn’t cure the main cause of bladder dysfunction. Therefore, structural and functional changes of the bladder may reappear over time after PRP therapy is stopped.

Response: Thanks for your professional recommendation. As suggested by the reviewer, the clinical application of PRP for bladder dysfunction has cited in the fifth paragraph of introduction section (please refer to page 4). Besides, we added bladder structural and functional changes after PRP therapy in animal model in the fourth paragraph of introduction section (please refer to page 4).

We fully agree with the opinion of the reviewer that “PRP therapy reduce bladder symptoms temporarily as it acts locally on the bladder. It doesn’t completely cure the main cause of OHD induced bladder dysfunction. Therefore, structural and functional changes of the bladder may reappear over time after PRP therapy is stopped. “ However, The purpose of the present study is to explore the treatment effect and molecular mechanism of PRP in postmenopausal rats with OAB induced by OVX. The use of OVX rats can provide a relevant model for studying postmenopausal OHD and its associated symptoms. The study design involved treating the rats with once-weekly PRP for four weeks following 12 months of OVX. The results showed that PRP therapy improved OAB symptoms, with more prominent improvements observed after four weeks of treatment. To determine how long the therapeutic effect of PRP may last, it is necessary to design another long-term follow-up experiments. Long-term studies could help to determine the duration of PRP's therapeutic effect on OAB symptoms and whether any additional treatments or interventions may be necessary to sustain these effects over time.

  • Results: Data presentation is the weakest part of this article. The data, as presented, is somewhat unclear and difficult. Unnecessary details are provided while crucial findings are not described clearly and remain obscure. The data, as presented, suggest that PRP can reverse almost all structural and functional changes of the bladder induced by OHD in the OVX model, which is somewhat surprising.

Response: Thanks for your recommendation. We have revised and presented our results in the Results section.

  • Results: As stated earlier, OHD is the main cause of bladder dysfunction in the OVX model and PRP has no significant effect on OHD. Thus, PRP may somewhat improve OHD-induced bladder symptoms but may not be capable of normalizing all functional, structural, and molecular changes in the OVX model to the control levels. Please reanalyze the data or provide clarifications.

Response: As mentioned by the reviewer, PRP therapy is aimed at alleviating bladder symptoms caused by OHD while having no effect on OHD itself which is the main cause of bladder dysfunction in the OVX model. Our study found that OVX-treated rats exhibited significant bladder hyperactivity, as evidenced by increased micturition frequency and decreased bladder capacity, along with defective urothelial mucosa, interstitial fibrosis, and collagen accumulation. The RAS/ERK1/2 and NFƘB/COX-2 signaling pathways exacerbated bladder damage and fibrosis. However, PRP instillation for four weeks regulated inflammatory fibrotic biosynthesis, promoted cell proliferation, matrix synthesis, and mucosal regeneration through the HA/PI3K/AKT/SOX-9 signaling pathway. PRP also enhanced angiogenesis and ECM synthesis through the VEGF/VEGF-R and PI3K/AKT/e-NOS signaling pathways for bladder repair. Thus, our findings suggest that PRP instillation improves OHD-induced bladder dysfunction by ameliorating inflammation, promoting cell proliferation, and enhancing mucosal regeneration and angiogenesis. PRP also promotes bladder regeneration and repair by modulating fibroblast-myofibroblast transition and increasing ECM synthesis.

  • Results: The data in table 1 should be described clearly. Please provide a legend for table 1 and provide a clear description of the findings. Under urodynamic data, please clarify which data is from metabolic cage studies and which data is from CMG?

Response: We described the data of Table 1 in the Result section on page 7, as “Bladder function was evaluated urodynamic parameters by cystometrography (CMG) and voiding behavior by metabolic cage. The urodynamic parameters, such as peak micturition pressure, micturition frequency, micturition interval, voided volume and non-voided contraction. Besides, tracing analysis of voiding behavior by metabolic cage revealed that.. voiding volume and …micturition frequency. …………“

  • Results: What was the reason for measuring lipid profile (cholesterol, triglycerides, LDL, HDL)? What is the link between OHD, PRP therapy, and lipid profile? Did PRP therapy significantly change lipid profile? If so, why?

Response: Physical and biochemical parameters were analyzed to eliminate interference factors of OAB, such as metabolic syndrome, hypertension, diabetes mellitus, hyperlipidemia, renal disease, and liver disease. Additionally, postmenopausal females may experience a modest increase in LDL cholesterol levels, and either no change or a slight decrease in HDL cholesterol levels due to the loss of estrogen. However, estrogen administration has been found to decrease LDL cholesterol levels while increasing triglycerides and HDL cholesterol levels, according to a study by Kenneth R. Feingold, Eliot A. Brinton, et al. (2020).

After 12 months of bilaterally OVX surgery, our data showed that rats developed metabolic syndrome, as evidenced by changes in physical indicators such as serum estradiol concentration, waist circumference, body weight, bladder weight, and the ratio of bladder weight (mg)/body weight (g), as well as in serum and biochemical parameters including GOT, GPT, triglycerides, cholesterol, LDL, glucose, insulin, and LDH. Compared to the sham group, the OVX group, OVX+PRP group, and OVX+PPP group had significantly elevated serum parameters associated with metabolic syndrome, with the exception of HDL and insulin. However, treatment with PRP and PPP had no significant effect on the physical characteristics and serum parameters in the OVX model, except for bladder weight and the ratio of bladder weight/body weight, and did not significantly alter the lipid profile. These results indicated that OHD in combination with metabolic abnormalities caused a profound negative effect on lipid profile. The OVX treated with PRP and PPP groups had the limitation in restoring the control level (Please refer to the Result section on page 5).

  • Results: It seems that PPP significantly diminished voiding frequency, voided volume, peak micturition pressure, and number of non-voiding contractions. The data in table 1 suggest that PRP and PPP had similar effects on the urodynamic parameters. Please provide clarifications and describe how PPP produced significant effects through which downstream mechanisms.

Response: As suggested by the reviewer, the following information has been added to the Discussion section on pages 29 and 30, as “Based on the data of urodynamic parameter and voiding behavior, both PRP and PPP treatment significantly improved urodynamic parameters compared to the OVX group in Table 1 and Figure 1. According to bladder pathological features by immunostaining and western blot, the expression of inflammatory and fibrosis markers [TGF-ß1, fibronectin, collagen I, COX-2 and NFƘB-p65] were enhanced in the OVX group in comparison with the sham group, however, the expression was declined in the OVX + PRP group versus the OVX + PPP group. Moreover, the expressions of adhesion protein (E-Cadherin), differentiated urothelial marker (UPKIII), proliferation marker (Ki-67 and CK14), tight junction proteins (Claudin-4 and ZO-1), angiogenesis related proteins and receptors (α-SMA and VEGF) and HA receptors (CD44 and RHAMM) was reduced in the OVX group compared to the sham group, however, was increased by in the OVX + PRP group versus in the OVX + PPP group. From the above comprehensive results, OHD after bilaterally OVX resulted in urothelial atrophy due to less expression of urothelial structure, exacerbated urothelial lining defects and reduced angiogenic remodeling, resulting in bladder damage. Nevertheless, the effects of weekly PRP and PPP instillation modulated the fibrotic biosynthesis, improved urothelial barrier, increase bladder regeneration and angiogenic potential to ameliorate bladder injury in Figures 2-4. However, the effect of PRP treatment was more potential than PPP. Although PPP was a plasma fraction that is depleted of platelets, it still contains several cytokines, growth factors and serum proteins.”

In a study by Kobayashi et al. (2015), the number of platelets in freshly prepared PRP and PPP samples was determined using an automated hematology analyzer (as shown in Figure R1). Compared to PPP, PRP preparations showed significantly higher concentrations of both VEGF and PDGF. PRP was found to be more effective in promoting wound closure in HUVEC cultures using a scratch assay. In the chicken chorioallantoic membrane assay (as shown in Figure R2 and R3), PRP was found to be more potent than PPP in α-SMA staining, which is used to estimate the number of mature blood vessels. Western blotting in HUVEC cultures also showed increased phosphorylation of VEGFR2 in the PRP group compared to the PPP group. This is likely due to the direct action of TGF-β, PDGF, and VEGF, which are all concentrated in PRP preparations and capable of stimulating fibroblast proliferation. Previous literature has also shown that PRP stimulated VEGF and VEGF receptor to promote endothelial cell motility and wound repair, and that PDGF and VEGF synergistically function to facilitate neovascularization during the wound healing process. Additionally, PDGF stimulates the chemotaxis of macrophages and neutrophils and enhances the secretion of TGF-β from macrophages. Based on these findings, the authors suggest that high concentrations of VEGF, TGF-β, and PDGF in PRP cooperatively induce reciprocal interactions between bladder cells and ECM, leading to improved neovascularization and increased blood supply and nutrients influx necessary for cell regeneration in damaged tissue.

Figure R1. Preparation of PPP and PRP samples. (Kobayashi, Kawase et al. 2015)

Figure R2. The effects of clotted PPP, clotted PRP, and PRF membrane preparations on new blood vessel formation in the CAM assay (Kobayashi, Kawase et al. 2015).

Figure R3. Immunohistochemical examination of the effects of PPP and PRP preparations on formation of α-SMA+ matured blood vessels in the CAM (Kobayashi, Kawase et al. 2015).

  • Results: Table 1 shows that PPP improved voided volume but the CMG data in figure 1 shows the opposite. Did PPP improve both voided volume and CMG changes? Please provide clarifications and describe how PPP acted on the dysfunctional bladder and improved urodynamic changes.

Response: PPP treatment improved bladder voided volume but decreased voiding frequency as shown in Table 1 and Figure 1. Table 1 showed the mean, standard deviation (SD), and p values of urodynamic parameters for 10 rats in each group. The average values of OVX+PPP group could partial reverse the bladder dysfunction but not completely normalized to the levels recorded in the sham group. The representative urodynamic picture was not exactly the same as the average value. Our data showed PPP improved both voided volume and frequency compared to the OVX group. PPP is a plasma fraction that is depleted of platelets. However, it still contains several cytokines, growth factors and serum proteins. In PRP preparations, both VEGF and PDGF were significantly more concentrated than PPP. The previous study showed that PRP had more effect on the neovascularization than PPP (Kobayashi, Kawase et al. 2015). This phenomenon could be explained by the effect of TGF-β, PDGF and VEGF, all of which are concentrated in both PRP and PPP preparations and capable of stimulating the proliferation of fibroblasts.

  • Results: Figure 1 suggests that PRP therapy completely normalized bladder pressure, frequency of contractions, and voided volume of the OVX group to the levels recorded in sham controls. The effects of PRP shown in figure 1 are dramatic and somewhat surprising. Did PRP cure all functional changes in the OVX group and normalized them to the control levels? Downstream mechanisms mediating such a dramatic effect of PRP should be described clearly.

Response:  Figure 1A is a representative figure for cystometric study and voiding pattern. Table 1 showed the mean, standard deviation (SD), and p values of urodynamic parameters for 10 rats in each group. The average values of OVX+PRP group could partial reverse the bladder dysfunction but not completely normalized to the levels recorded in the sham group.

PRP is a complex mixture of various growth factors, cytokines, and other bioactive molecules that can promote tissue repair and regeneration. While the exact mechanism of action of PRP on bladder repair is not yet fully understood, previous studies have suggested that PRP can stimulate the production and release of various growth factors and cytokines that promote angiogenesis, cell proliferation, and tissue regeneration. One of the key growth factors that PRP can stimulate is vascular endothelial growth factor (VEGF), which plays a critical role in promoting the formation of new blood vessels (angiogenesis) and facilitating the migration and proliferation of endothelial cells. PRP can also stimulate the production and release of platelet-derived growth factor (PDGF), which can enhance the chemotaxis of macrophages and neutrophils, as well as the secretion of transforming growth factor-beta (TGF-β) from macrophages (Barrientos, Stojadinovic et al. 2008).

It is important to note that the molecular mechanism of PRP on bladder repair is likely to be complex and involve multiple signaling pathways. Therefore, future studies using PRP-treated primary cell cultures (in vitro) and inhibitors (in vivo) will be needed to more clearly elucidate the exact mechanism of action of PRP on bladder repair. Besides, we have added the paragraph in the Discussion section on pages 29-32.

  • Results: Structural data also suggest that PRP therapy reversed almost all structural changes of the OVX group to the control levels. However, the data as presented is not clear and need clarifications.

Response: Thanks for your recommendation. We have revised and presented our result in the Result section and Figure 2 on pages 9-12 for “Therapeutic effect of PRP improved OHD-induced pathological alteration by Masson’s trichrome staining, immunostaining and western blots.” Additionally, We have revised and presented our results in the Result section and Figure 3 on pages 12-15 for Effects of PRP instillation strengthened proliferation and tight junction reconstruction.

  • Results: The trichrome, immunofluorescence, and western blotting data in figure 2 do not show the findings clearly. The data is poorly presented. The descriptions provided in figure legend cannot be found in the corresponding sections of figure 2.

Response: Thanks for your recommendation. We have revised and adjusted the resolution in Figure 2 in the Result section on page 11. Besides, we performed the description of figure legend in the Result section on page 12.

  • Results: Same issue with figure 3. Unclear and poorly defined figures are presented. The figures do not show the findings clearly. Detailed descriptions made in figure legend cannot be found in the figures.

Response: Thanks for your recommendation. We have revised and adjusted the resolution in Figure 3 in the Result section on page 14. Besides, we performed the description of figure legend in the Result section on page 15.

  • Results: Sections A-L of figure 4 do not show the findings clearly. No distinction is made between arterioles, venules, and lymphatic ducts. It seems that all those structures are interpreted as arterioles. Sections M and O are blurry and unclear.

Response: To evaluate bladder angiogenic potential, the expression of various angiogenesis markers was analyzed using immunostaining and western blotting analysis. Specifically, the number of mature blood vessels was determined based on α-SMA immunostaining, a marker of vascular smooth muscle cells, except lymphatic ducts. However, the morphology of small artery, small vein, arterioles and venules was sometimes difficult to distinguish under α-SMA and laminin immunostaining. Hence, the level of bladder angiogenesis markers was quantified using western blotting analysis of α-SMA, laminin, and VEGF expression. The results showed that α-SMA immunostaining was widely distributed in myofibroblasts and smooth muscle of micro-vessels beneath the urothelial basal layer, in the lamina propria and muscularis layers of the sham group (please refer to the Result section on page 17).

We have also revised and adjusted the resolution in Figure 4M and 4O in the Result section on page 17.

  • Results: The immunofluorescence data in figure 6 do not show the findings clearly. Detailed descriptions made in figure 6 legend cannot be found in the figures.

Response: Thanks for your recommendation. We have revised and adjusted the resolution in Figure 6. Besides, we revised legend for figure 6 on page 22.

  • Results: Sections A, B, C, and D of figure 7 do not show the data clearly. It is not clear what the authors intend to show in the fluorescence section of this figure.

Response: Thanks for your recommendation. We have revised and adjusted the resolution in Figure 7 on page 24. Besides, we revised legend for figure 7 in the Result section on pages 24 and 25.

  • Results: Figure 8A is blurry and unclear.

Response: We have adjusted the resolution in Figure 8 in the Result section on page 26.

  • Results: Sections B, C D and E of figure 9 are not needed and should be removed.

Response: As suggested by the reviewer, we have deleted sections B, C D and E of figure 9 in the Result section on page 28.

  • Discussion: The authors should provide a comprehensive and analytical discussion of the findings and clarify how PRP therapy produced such dramatic effects and completely normalized all structural and functional changes of the OVX group to the levels of sham control group while the main cause of bladder dysfunction (OHD) remained untreated

Response: Thanks for your recommendation. We have reorganized and performed a comprehensive and analytical discussion of how PRP therapy produced pronounce therapeutic effects in the Discussion section on pages 29-33.

  • Discussion: The discussion lacks cohesion as to how the findings presented in this paper coalesce into new information pertaining to PRP therapy and bladder dysfunction.

Response: Thanks for your recommendation. We have integrated, discussed and presented the Discussion section on pages 29-33.

  • Discussion: Some paragraphs of the discussion are more appropriate for the introduction, wherein it justifies the experiments performed but does not place the new findings in context of what is already known about PRP therapy for bladder dysfunction.

Response: Thanks for your recommendation. We have integrated, discussed and presented the Discussion section (pages 29-33) and moved some paragraphs to the Introduction section.

  • Discussion: A deeper discussion that places the new findings in a broader context whilst offering alternative interpretations of the data may enhance the quality of this article.

Response: Thanks for your recommendation. We have integrated, discussed and presented the Discussion section on pages 29-33.

  • Materials and Methods: The materials and method section is well done.

Response: Thank you for the compliment.

  • Materials and Methods: Figure 10 in the method section is not needed and should be removed.

Response: As suggested by the reviewer, we removed figure 10 in the Method section on page 34.

These references are listed here:

  1. Feingold KR, Brinton EA, Grunfeld C. The Effect of Endocrine Disorders on Lipids and Lipoproteins. [Updated 2020 Mar 9]. In: Feingold KR, Anawalt B, Blackman MR, et al., editors. Endotext [Internet]. South Dartmouth (MA): MDText.com, Inc.; 2000-. Available from: https://www.ncbi.nlm.nih.gov/books/NBK409608/
  2. Mito Kobayashi, Tomoyuki Kawase, Kazuhiro Okuda, Larry F. Wolff and Hiromasa Yoshie. In vitro immunological and biological evaluations of the angiogenic potential of platelet-rich fibrin preparations: a standardized comparison with PRP preparations. International Journal of Implant Dentistry (2015) 1:31.
  3. Richardson TP, Peters MC, Ennett AB, Mooney DJ. Polymeric system for dual growth factor delivery. Nat Biotechnol. 2001;19:1029–34.
  4. Barrientos S, Stojadinovic O, Golinko MS, BremH, Tomic-CanicM. Growth factors and cytokines in wound healing. Wound Repair Regen. 2008;16:585–601.

Reviewer 2 Report

It is an interesting article, however, it is lengthy and need to be re-organized.

In addition, some concern should be cleared.

1.      Title “Effects of therapeutic platelet-rich plasma on overactive bladder via modulating hyaluronan synthesis in ovariectomized rat.”

Please showed clinical evidence that OAB is related to deficiency of hyaluronan synthesis. It seemed to be related to interstitial cystitis, which is different from OAB.

 OAB is different from IC, as below reference.

Comparing concentration of urinary inflammatory cytokines in interstitial cystitis, overactive bladder, urinary tract infection, and bladder cancer
Michael B Chancellor, Laura E Lamb, Elijah P Ward, Sarah N Bartolone, Alexander Carabulea, Prasun Sharma, Joseph Janicki, Christopher Smith, Melissa Laudano, Nitya Abraham, Bernadette M M. Zwaans Urological Science, Year 2022, Volume 33, Issue 4 [p. 199-204]
DOI: 10.4103/UROS.UROS_26_22

2.      Introduction- lengthy and redundant description, please cut in precise.

For example, In the ER-ß-/- 64 female mice, the pathological morphology of urothelial ulceration, atrophy and bladder 65 hyperactivity were shown to be compatible with interstitial cystitis and bladder pain syn- 66 drome (IC/BPS) in human [9]. OAB is different from IC in clinical symptoms and pathopysiologies.

3.      Moreover, Botox injection and tibial nerve stimulation for bladder were the second line treatment owing to invasive procedure with side effects: increasing residual urine, catheter drainage risk, hematuria, nerve pain and nerve injury.

Please revised botox as 3rd line treatment for OAB.

4.      In view of the importance of HA metabolism in maintaining the integrity of bladder structure, function, and tissue repair, the current study hypothesized that PRP treatment might modulate the expression of HA-metabolizing enzymes and receptors for tissue remodeling, thereby improving bladder repair in OHD-induced OAB.

Please clear “OHD-induced OAB or IC”, bladder mucosa defect is related to IC, but not OAB.

5.      This work intended to develop a non-pharmaceutical PRP therapy directly from autologous blood without the risk of rejection or allergy.

The authors claimed “non-pharmaceutical PRP therapy”. I was confused, PRP is full  of  growth and cytokine, why not pharmaceutical ?

6.      However, the bladder weight and the ratio of bladder weight / body weight of the OVX group were significantly reduced than those of the sham group. Any evidence of this finding “ reduced bladder weight / body weight” related to human OAB?

7.      . In contrast, both the OVX + PRP group and the OVX + PPP group showed significantly reduced peak micturition pressure as well as micturition frequency, and increased bladder capacity as compared with the OVX group. Why PRP and PPP have similar urodynamic effects?

8.      Result section- lengthy and redundant description, please cut in precise.

9.      Taken together, the above findings implied that that the OVX-treated rats exhibited significant bladder hyperactivity, abnormal detrusor activity with an increase in micturition frequency and deteriorated bladder capacity, whereas PRP treatment significantly improved bladder capacity and ameliorated OAB induced by OHD. Please delete or replace, not good in the result section.

10.  According to the above data, the inflammatory and fibrosis markers were noticeably decreased in the OVX + PRP group compared to the OVX group. Moreover, PRP might play an important role in OHD-induced OAB recovery by inhibiting the expression of NFkB and TGF-β. Please delete or replace, not good in the result section.

11.  Based on the morphological evaluation and western blot findings, OHD after bilat erally OVX resulted in urothelial atrophy due to less expression of urothelial structure markers, which promoted bladder fibrotic biosynthesis, exacerbated urothelial lining defects in interstitial fibrosis, resulting in bladder damage. Nevertheless, the effects of weekly PRP and PPP instillation modulated the fibrotic biosynthesis, improved urothelial barrier and ameliorated bladder injury. Please delete or replace, not good in the result section.

12.  Moreover, the OVX-treated rats decreased the expressions of cell proliferation as well as angiogenesis, and altered HA production, which might attribute to OAB. Please show clinical evidence. It is not a right hypothesis for OAB.

13.  “Besides, in gentamicin-induced nephrotoxicity rat 648 model, PRP secreted growth factors could promote renal epithelial proliferation and improve renal parenchymal fibrosis [40]” – not related to current study.

14.  These findings suggested that intravesical instillation of PRP could recruit fibroblast, alternate HA synthesis and improve such OHD- 675 induced OAB through increasing HA expression. Did you suggest PRP instillation in human study, instead of injection?

15.  However, PRP and PPP instillation could recruit fibroblast, alternate HA synthesis and improve OAB through increasing HA expression. Any evidence to show HA abnormality in human OAB study?

16.  Previous studies have shown at least a five-fold increase in PRP concentration could achieve therapeutic effect.  Not clear, please cite reference.

Author Response

Reviewer #2:

It is an interesting article, however, it is lengthy and need to be re-organized.

In addition, some concern should be cleared.

Comments:

  • Title “Effects of therapeutic platelet-rich plasma on overactive bladder via modulating hyaluronan synthesis in ovariectomized rat.”
  1. Please showed clinical evidence that OAB is related to deficiency of hyaluronan synthesis. It seemed to be related to interstitial cystitis, which is different from OAB. OAB is different from IC, as below reference.
  2. Comparing concentration of urinary inflammatory cytokines in interstitial cystitis, overactive bladder, urinary tract infection, and bladder cancer

Michael B Chancellor, Laura E Lamb, Elijah P Ward, Sarah N Bartolone, Alexander Carabulea, Prasun Sharma, Joseph Janicki, Christopher Smith, Melissa Laudano, Nitya Abraham, Bernadette M M. Zwaans Urological Science, Year 2022, Volume 33, Issue 4 [p. 199-204] DOI: 10.4103/UROS.

Response: Thanks for your professional recommendation.

  1. Currently, there is insufficient clinical experimental evidence to establish a direct link between menopause-induced OAB and the deficiency of hyaluronan synthesis. Moreover, it is challenging to analyze the role of HA through immunostaining and western blotting since bladder tissue is not readily accessible in clinical trials. Hence, we have to rely on rat models to verify the correlation between HA and OAB. As you mentioned, some clinical trials have indicated that HA can alleviate symptoms of bladder overactivity induced by interstitial cystitis (IC), but the underlying mechanism remains unclear (Ho et al., 2011). In addition, the therapeutic effect of HA and its potential mechanism of action have been reported in the ketamine-induced ulcerative cystitis (KIC) rat model. Therefore, PRP, which contains growth factors that enhance the expression of HA receptors and hyaluronan synthases (HAS) enzymes, while reducing hyaluronidases (HYALs), could be an effective treatment for OHD-induced OAB (Lee et al., 2017).
  2. Chancellor et al. discovered that the concentrations of GRO and MCP-1 in urine were higher in individuals with OAB wet compared to healthy controls, as determined by a multiplex Luminex assay. However, these four cytokines (GRO, IL-6, IL-8, and MCP-1) were specifically selected as they had been previously identified as potential biomarkers for the diagnosis of IC, rather than OAB. This study did not examine the impact of urinary cytokines and chemokines on pathological features and mechanisms (Chancellor et al., 2022). The study compared the concentrations of urinary inflammatory cytokines in interstitial cystitis, overactive bladder, urinary tract infection, and bladder cancer, which can be used as a reference for designing OAB experiments in the future.

Table 3: Concentration of urine inflammatory cytokines in study groups. (Chancellor et al. 2022)

  • Introduction- lengthy and redundant description, please cut in precise.

For example, In the ER-ß-/- female mice, the pathological morphology of urothelial ulceration, atrophy and bladder hyperactivity were shown to be compatible with interstitial cystitis and bladder pain syndrome (IC/BPS) in human [9]. OAB is different from IC in clinical symptoms and pathopysiologies.

Response: Thanks for your recommendation. As suggested by the reviewer, we have revised the redundant description in the Introduction section on pages 2-3.

  • Moreover, Botox injection and tibial nerve stimulation for bladder were the second line treatment owing to invasive procedure with side effects: increasing residual urine, catheter drainage risk, hematuria, nerve pain and nerve injury.

Please revised botox as 3rd line treatment for OAB.

Response: As suggested by the reviewer, we have revised the description of the Introduction section (please refer to page 2, line 77).

  • In view of the importance of HA metabolism in maintaining the integrity of bladder structure, function, and tissue repair, the current study hypothesized that PRP treatment might modulate the expression of HA-metabolizing enzymes and receptors for tissue remodeling, thereby improving bladder repair in OHD-induced OAB.

Please clear “OHD-induced OAB or IC”, bladder mucosa defect is related to IC, but not OAB.

Response: As suggested by the reviewer, we have revised the description of the Introduction section on page 4.

  • This work intended to develop a non-pharmaceutical PRP therapy directly from autologous blood without the risk of rejection or allergy.

The authors claimed “non-pharmaceutical PRP therapy”. I was confused, PRP is full  of  growth and cytokine, why not pharmaceutical ?

Response: Platelet-rich plasma (PRP) is a concentrated solution of growth factors derived from a patient's own blood that is used to aid in the natural healing process. According to the FDA's definition, PRP is technically considered a "biologic" material, rather than a medication, and is therefore typically classified as a cellular therapy. PRP therapy has been approved by the government agency for clinical patients (Jones, Togashi et al. 2018, Ennis 2019). Human clinical studies have shown that PRP treatment has a beneficial effect on cell proliferation and collagen production, as well as the stimulation of matrix-degrading enzymes (matrix metalloproteinases) in tenocytes, alveolar bone cells, osteoblasts, fibroblasts, and bone marrow stem cells. Clinical applications of PRP in the urinary system have demonstrated its effectiveness in treating stress urinary incontinence, recurrent bacterial cystitis, erectile dysfunction, and interstitial cystitis/bladder pain syndrome (IC/BPS). In the present study, PRP derived from animals with similar body composition was used, and it contained various growth factors that modulated HA production for comprehensive bladder repair. A description of this is provided in the Introduction section on pages 3 and 4.

  • However, the bladder weight and the ratio of bladder weight / body weight of the OVX group were significantly reduced than those of the sham group. Any evidence of this finding “ reduced bladder weight / body weight” related to human OAB?

Response: Previous data indicated that OAB were caused by various diseases and were not directly correlated with the ratio of bladder weight to body weight. However, it is not feasible to obtain human bladder specimens for this purpose. In this study, we found that in OHD-induced OAB, the bladder weight in the OVX group was lower than that in the sham group, despite the OVX group having a higher body weight. Therefore, the ratio of bladder weight to body weight in the OVX group was lower than that in the sham group (as shown in Table underneath).     

Comparing with the bladder weight, bladder weight and the ratio of bladder weight /body weight.

Control group

Ketamine group

Sham group

OVX group

Body weight (g)

267.8 ± 27.2

263.6 ± 36.8

345.2 ± 28.6

469. 8 ± 41.4**

Bladder weight (mg)

166.0 ± 12.6

188.0 ± 29.8**

218.0 ± 16.6

196.7 ± 25.8*

The ratio of bladder weight (mg)/body weight (g)

0.62 ± 0.06

0.71 ± 0.08*

 0.60 ± 0.09

 0.41 ± 0.06**

Experimental time

3 months

12 months

Reference

Lee et al., 2017.

Present study

  • In contrast, both the OVX + PRP group and the OVX + PPP group showed significantly reduced peak micturition pressure as well as micturition frequency, and increased bladder capacity as compared with the OVX group. Why PRP and PPP have similar urodynamic effects?

Response: Based on the data of urodynamic parameter and voiding behavior, both PRP and PPP treatment significantly improved urodynamic parameters compared to the OVX group as shown in Table 1 and Figure 1. Although PRP and PPP have similar effects, the effect of PRP treatment was found to be more potent than that of PPP, likely due to the higher concentration of growth factors and other bioactive molecules in PRP. Based on immunostaining and western blot analysis of bladder pathology, the expression of inflammatory and fibrosis markers such as TGF-ß1, fibronectin, collagen I, COX-2, and NFkB-p65 were higher in the OVX group than in the sham group. However, their expression was reduced in the OVX + PRP group compared to the OVX + PPP group. Additionally, the expressions of adhesion protein (E-cadherin), differentiated urothelial marker (UPKIII), proliferation markers (Ki-67 and CK14), tight junction proteins (Claudin-4 and ZO-1), angiogenesis-related proteins and receptors (α-SMA and VEGF), and HA receptors (CD44 and RHAMM) were lower in the OVX group than in the sham group. However, their expression was increased in the OVX + PRP group compared to the OVX + PPP group. These findings suggest that bilateral OVX resulted in urothelial atrophy due to decreased expression of urothelial structure, increased urothelial lining defects, and reduced angiogenic remodeling, ultimately leading to bladder damage. Nevertheless, weekly PRP and PPP instillation had a positive effect on modulating fibrotic biosynthesis, improving urothelial barrier, increasing bladder regeneration, and promoting angiogenic potential to alleviate bladder injury, as shown in Figures 2-4. However, the PRP treatment had a greater potential effect than PPP, even though PPP is a plasma fraction that lacks platelets but still contains several cytokines, growth factors, and serum proteins.

Kobayashi et al. (Kobayashi, Kawase et al. 2015) demonstrated that PRP contains higher concentrations of VEGF and PDGF compared to PPP. In vitro studies using HUVEC cultures showed that PRP was the most effective in promoting wound closure in the scratch assay. In the chicken chorioallantoic membrane assay, PRP had a stronger staining potency for estimating the number of mature blood vessels compared to PPP. Furthermore, western blot analysis of HUVEC cultures showed that the phosphorylation of VEGFR2 was increased in the PRP group compared to the PPP group. This may be due to the direct action of TGF-β, PDGF, and VEGF, which are all highly concentrated in PRP and capable of stimulating the proliferation of fibroblasts.

Previous study (Richardson, Peters et al. 2001) has also reported that PRP stimulated VEGF and VEGF receptor to enhance endothelial cell motility and wound repair, while PDGF and VEGF work together to promote neovascularization during the wound healing process. In addition, PDGF stimulates the chemotaxis of macrophages and neutrophils and enhances the secretion of TGF-β from macrophages (Barrientos, Stojadinovic et al. 2008). It is suggested that the high concentrations of VEGF, TGF-β, and PDGF provided by PRP cooperatively induce reciprocal interactions between bladder cells and ECM, leading to improved neovascularization, increased blood supply, and influx of necessary nutrients for cell regeneration in damaged tissue.

  • Result section- lengthy and redundant description, please cut in precise.

Response: As suggested by the reviewer, we have revised the description of the Result section to make it more clear and concise.

  • Taken together, the above findings implied that the OVX-treated rats exhibited significant bladder hyperactivity, abnormal detrusor activity with an increase in micturition frequency and deteriorated bladder capacity, whereas PRP treatment significantly improved bladder capacity and ameliorated OAB induced by OHD. Please delete or replace, not good in the result section.

Response: As suggested by the reviewer, we have revised the description of the Result section (please refer to page 7).

  • According to the above data, the inflammatory and fibrosis markers were noticeably decreased in the OVX + PRP group compared to the OVX group. Moreover, PRP might play an important role in OHD-induced OAB recovery by inhibiting the expression of NFkB and TGF-β. Please delete or replace, not good in the result section.

Response: As suggested by the reviewer, we have revised the description of the Result section (please refer to page 10).

  • Based on the morphological evaluation and western blot findings, OHD after bilaterally OVX resulted in urothelial atrophy due to less expression of urothelial structure markers, which promoted bladder fibrotic biosynthesis, exacerbated urothelial lining defects in interstitial fibrosis, resulting in bladder damage. Nevertheless, the effects of weekly PRP and PPP instillation modulated the fibrotic biosynthesis, improved urothelial barrier and ameliorated bladder injury. Please delete or replace, not good in the result section.

Response: As suggested by the reviewer, we have revised the description of the Result section (please refer to page 10,).

  • Moreover, the OVX-treated rats decreased the expressions of cell proliferation as well as angiogenesis, and altered HA production, which might attribute to OAB. Please show clinical evidence. It is not a right hypothesis for OAB.

Response: Overactive bladder (OAB) is characterized by bothersome lower urinary tract symptoms of unknown etiology. Current understanding of the pathophysiology of these bothersome symptoms indicates that there is a contribution from both neurogenic, urothelial and myogenic sources. In addition, it is now thought that the urothelium plays a more active role in bladder function than simply being a barrier ( Pradeep Tyagi, 2011).

Simplified diagram of these potential contributing mechanisms. This short review focuses on the local pathophysiology ( Pradeep Tyagi, 2011).

Overactive bladder symptoms become more prevalent during menopause and worsen with increasing vaginal atrophy. The physical changes in the bladder trigone correspond with the epidemiological increase in bladder symptoms and cystitis. The decline in serum estrogen concentration in postmenopausal women causes the postmenopausal genitourinary syndrome, which includes symptoms such as vaginal atrophy. Estrogen therapy can stimulate the synthesis of hyaluronic acid (Uzuka, Nakajima et al. 1981), which has efficacy similar to vaginal estrogens for treating the signs of vaginal atrophy and dyspareunia (Dos Santos, Uggioni et al. 2021). OVX-treated rats showed decreased expression of cell proliferation and angiogenesis-associated protein, as well as altered HA production, which may contribute to OAB.

  • “Besides, in gentamicin-induced nephrotoxicity rat model, PRP secreted growth factors could promote renal epithelial proliferation and improve renal parenchymal fibrosis [40]” – not related to current study.

Response: As suggested by the reviewer, we have deleted this statement from the Introduction section (please refer to page 3).

  • These findings suggested that intravesical instillation of PRP could recruit fibroblast, alternate HA synthesis and improve such OHD- induced OAB through increasing HA expression. Did you suggest PRP instillation in human study, instead of injection?

Response: The optimal route of PRP administration for OAB treatment remains to be determined and may depend on various factors, including the specific goals of therapy and individual patient characteristics. PRP injection has been investigated in some studies, and clinical applications of PRP in the urinary system have been demonstrated for various conditions such as IC, stress urinary incontinence, and recurrent bacterial cystitis. However, due to the small blood volume of rats, injecting PRP with autologous blood can cause death, while allogeneic blood injection can cause rejection. Therefore, we used PRP instillation in rats instead. While PRP therapy has shown potential in treating IC/BPS symptoms and enhancing functional bladder capacity, its potential effect on treating OAB symptoms in menopausal women and its underlying mechanism are still unclear.

  • However, PRP and PPP instillation could recruit fibroblast, alternate HA synthesis and improve OAB through increasing HA expression. Any evidence to show HA abnormality in human OAB study?

Response: HA has been shown to have similar efficacy to vaginal estrogens for treating vaginal atrophy and dyspareunia, and its altered production may contribute to OAB in OVX-treated rats. Additionally, HA has been shown to play a role in tissue repair, wound healing, and angiogenesis. Intravesical instillation of HA has also been shown to alleviate bladder ulcerative symptoms.

It is possible that PRP therapy could modulate the expression of HA-metabolizing enzymes and receptors for tissue remodeling, leading to improved bladder repair in OHD-induced OAB. While there is no evidence of HA abnormalities in human OAB studies, previous research has shown that HA is involved in tissue repair, wound healing, and angiogenesis, and has been effective in treating bladder ulcerative symptoms and cyclophosphamide-related bladder overactivity. PRP, which contains various growth factors that can modulate HA production, may enhance intracellular TGF-β1 expression, further regulating HA production. Therefore, it is possible that PRP therapy could be a potential treatment option for OAB in menopausal women, though further research is needed to confirm its efficacy and underlying mechanisms.

  • Previous studies have shown at least a five-fold increase in PRP concentration could achieve therapeutic effect. Not clear, please cite reference.

Response: As suggested by the reviewer, we have cited references in the seventh paragraph of the Discussion section, as “Previous studies have shown at least a five-fold increase in PRP concentration could achieve therapeutic effect (Sadoghi, Lohberger et al. 2013, Haunschild, Huddleston et al. 2020).” (please refer to page 32).

These references are listed here:

  • Michael B Chancellor, Laura E Lamb, Elijah P Ward, Sarah N Bartolone, Alexander Carabulea, Prasun Sharma, Joseph Janicki, Christopher Smith, Melissa Laudano, Nitya Abraham, Bernadette M M. Zwaans Urological Science, Year 2022, Volume 33, Issue 4 [p. 199-204] DOI: 10.4103/UROS.
  • Yi-Lun Lee, Kun-Ling Lin, Shu-Mien Chuang, Yung-Chin Lee, Mei-Chin Lu, Bin-Nan Wu, Wen-Jeng Wu, Shyng-Shiou F Yuan, Wan-Ting Ho, Yung-Shun Juan.  Elucidating Mechanisms of Bladder Repair after Hyaluronan Instillation in Ketamine-Induced Ulcerative Cystitis in Animal Model Am J Pathol . 2017 Sep;187(9):1945-1959.
  • Ho DR, Chen CS, Lin WY, Chang PJ, Huang YC. Effect of hyaluronic acid on urine nerve growth factor in cyclophosphamide-induced cystitis. Int J Urol. 2011 Jul;18(7):525-31.
  • Jones, I. A., R. C. Togashi and C. Thomas Vangsness, Jr. (2018). "The Economics and Regulation of PRP in the Evolving Field of Orthopedic Biologics." Curr Rev Musculoskelet Med 11(4): 558-565.
  • Ennis, T. (2019). "Is Platelet Rich Plasma Therapy Approved by FDA?" DELAWARE INTEGRATIVE HEALTHCARE Regenerative Medicine: https://deintegrativehealthcare.com/is-platelet-rich-plasma-therapy-approved-by-fda/.
  • Mito Kobayashi, Tomoyuki Kawase, Kazuhiro Okuda, Larry F. Wolff and Hiromasa Yoshie. In vitro immunological and biological evaluations of the angiogenic potential of platelet-rich fibrin preparations: a standardized comparison with PRP preparations. International Journal of Implant Dentistry (2015) 1:31.
  • Barrientos S, Stojadinovic O, Golinko MS, BremH, Tomic-CanicM. Growth factors and cytokines in wound healing. Wound Repair Regen. 2008;16:585–601.
  • Richardson TP, Peters MC, Ennett AB, Mooney DJ. Polymeric system for dual growth factor delivery. Nat Biotechnol. 2001;19:1029–34.
  • Pradeep Tyagi, Pathophysiology of the urothelium and detrusor. Cite as: Can Urol Assoc J 2011;5 (5Suppl2):S128-S130.
  • Uzuka, M., K. Nakajima, S. Ohta and Y. Mori (1981). "Induction of hyaluronic acid synthetase by estrogen in the mouse skin." Biochim Biophys Acta 673(4): 387-393.
  • Dos Santos CCM, Uggioni MLR, Colonetti T, Colonetti L, Grande AJ, Da Rosa MI.Hyaluronic Acid in Postmenopause Vaginal Atrophy: A Systematic Review. J Sex Med. 2021 Jan;18(1):156-166.
  • Grodstein F, Lifford K, Resnick NM, Curhan GC.Obstet Gynecol. Postmenopausal hormone therapy and risk of developing urinary 2004 Feb;103(2):254-60.
  • Hendrix SL, Cochrane BB, Nygaard IE, Handa VL, Barnabei VM, Iglesia C, Aragaki A, Naughton MJ, Wallace RB, McNeeley SG.Effects of estrogen with and without progestin on urinary incontinence. 2005 Feb 23;293(8):935-48.
  • Aya Nagaoka, Hiroyuki Yoshida, Sachiko Nakamura, Tomohiko Morikawa, Keigo Kawabata,Masaki Kobayashi, Shingo Sakai§, Yoshito Takahashi, Yasunori Okada, and Shintaro Inoue. Regulation of Hyaluronan (HA) Metabolism Mediated by HYBID (Hyaluronan-binding Protein Involved in HA Depolymerization, KIAA1199) and HA Synthases in Growth Factor-stimulated Fibroblasts*. HE JOURNAL OF BIOLOGICAL CHEMISTRY VOL. 290, NO. 52, pp. 30910 –30923
  • Francesco Trama, Ester Illiano, Alessandro Marchesi, Stefano Brancorsini, Felice Crocetto, Savio Domenico Pandolfo, Alessandro Zucchi and Elisabetta CostantiniUse of Intravesical Injections of Platelet-Rich Plasma for theTreatment of Bladder Pain Syndrome: A Comprehensive Literature Review. Antibiotics 2021, 10, 1194.
  • Yuan-Hong Jiang, Jia-Fong Jhang, Teng-Yi Lin, Han-Chen Ho, Yung-Hsiang Hsu and Hann-Chorng Ku Therapeutic Efficacy of Intravesical Platelet-Rich Plasma Injections for Interstitial Cystitis/Bladder Pain Syndrome—A Comparative Study of Different Injection Number, Additives and Concentrations. Frontiers in Pharmacology. 2022, 13, Article 853776.
  • Masuda H. Editorial comment to effect of hyaluronic acid on urine nerve growth factor in cyclophosphamide-induced cystitis. Int J Urol. 2011 Jul;18(7):531-2.
  • Haunschild, E. D., H. P. Huddleston, J. Chahla, R. Gilat, B. J. Cole and A. B. Yanke (2020). "Platelet-Rich Plasma Augmentation in Meniscal Repair Surgery: A Systematic Review of Comparative Studies." Arthroscopy: The Journal of Arthroscopic & Related Surgery 36(6): 1765-1774.
  • Sadoghi, P., B. Lohberger, B. Aigner, H. Kaltenegger, J. Friesenbichler, M. Wolf, T. Sununu, A. Leithner and P. Vavken (2013). "Effect of platelet-rich plasma on the biologic activity of the human rotator-cuff fibroblasts: A controlled in vitro study." J Orthop Res 31(8): 1249-1253.
  • Haunschild, E. D., H. P. Huddleston, J. Chahla, R. Gilat, B. J. Cole and A. B. Yanke (2020). "Platelet-Rich Plasma Augmentation in Meniscal Repair Surgery: A Systematic Review of Comparative Studies." Arthroscopy: The Journal of Arthroscopic & Related Surgery 36(6): 1765-1774.
  • Sadoghi, P., B. Lohberger, B. Aigner, H. Kaltenegger, J. Friesenbichler, M. Wolf, T. Sununu, A. Leithner and P. Vavken (2013). "Effect of platelet-rich plasma on the biologic activity of the human rotator-cuff fibroblasts: A controlled in vitro study." J Orthop Res 31(8): 1249-1253.

Round 2

Reviewer 1 Report

Thanks for your responses to my comments and the extensive revisions. Your revisions have significantly enhanced the quality of the manuscript.